



**Impacts of condensable particulate matter on atmospheric organic aerosols and fine**
**particulate matter (PM2.5) in China**
Mengying Li[1], Shaocai Yu[1+], Xue Chen[1], Zhen Li[1], Yibo Zhang[1], Zhe Song[1], Weiping Liu[1], Pengfei
Li[2+], Xiaoye Zhang[1,3], Meigen Zhang[4,5,6], Yele Sun[4,5], Zirui Liu[4,5], Caiping Sun[7], Jingkun Jiang[8,9],
Shuxiao Wang[8], Benjamin N. Murphy [10], Kiran Alapaty[10], Rohit Mathur[10], Daniel Rosenfeld[11], and John
H. Seinfeld[12]
[1]Research Center for Air Pollution and Health; Key Laboratory of Environmental Remediation and
Ecological Health, Ministry of Education, College of Environment and Resource Sciences, Zhejiang
University, Hangzhou, Zhejiang 310058, P.R. China
[2]College of Science and Technology, Hebei Agricultural University, Baoding, Hebei 071000, P.R. China
[3]Chinese Academy of Meteorological Sciences, China Meteorological Administration, Beijing 100081,
China.
[4]State Key Laboratory of Atmospheric Boundary Layer Physics and Atmospheric Chemistry (LAPC),
Institute of Atmospheric Physics (IAP), Chinese Academy of Sciences (CAS), Beijing 100029, China
[5]University of Chinese Academy of Sciences, Beijing 100049, China
[6]Center for Excellence in Urban Atmospheric Environment, Institute of Urban Environment, Chinese
Academy of Sciences, Xiamen, China
[7]Chinese Research Academy of Environmental Sciences, Beijing 100012, China
[8]State Key Joint Laboratory of Environment Simulation and Pollution Control, School of Environment,
Tsinghua University, Beijing 100084, China
[9]State Environmental Protection Key Laboratory of Sources and Control of Air Pollution Complex,
Beijing 100084, China
[10]Center for Environmental Measurement and Modeling, U.S. Environmental Protection Agency,
Research Triangle Park, NC 27711, USA
[11]Institute of Earth Sciences, The Hebrew University of Jerusalem, Jerusalem, Israel
[12]Division of Chemistry and Chemical Engineering, California Institute of Technology, Pasadena, CA
91125, USA.
*Correspondence to*: Shaocai Yu (shaocaiyu@zju.edu.cn); Pengfei Li (lpf_zju@163.com)





**Abstract**

Condensable particulate matter (CPM) emitted from stationary combustion and mobile sources
exhibits high emissions and a large proportion of organic components. However, CPM is not generally
measured when conducting emission surveys of PM in most countries, including China. Consequently,
previous emission inventories have not included emission rates for CPM. Here we construct an emission
inventory of CPM in China with a focus on organic aerosols (OA) based on collected CPM emission
information. Results show that OA emissions are enhanced twofold after the inclusion of CPM in a new
China inventory for the years 2014 and 2017. Considering organic CPM emissions and model
representations of secondary OA (SOA) formation from CPM, here a series of sensitivity cases have been
simulated using the three-dimensional Community Multiscale Air Quality (CMAQ) model to estimate the
contributions of CPM emissions to atmospheric OA and fine PM ($PM_{2.5}$) concentrations in China.
Compared with observations during a haze episode from October 14 to November 14, 2014, at a Beijing
site, estimates of temporal average primary OA (POA) and SOA concentrations are greatly improved after
including the CPM effects. These scenarios demonstrated the significant contributions of CPM emissions
from stationary combustion and mobile sources to POA (53 ~ 86%), SOA (48 ~ 67%), and total OA
concentrations (50 ~ 78%). Furthermore, contributions of CPM emissions to total OA concentrations were
demonstrated over the major 2+26 cities of Beijing-Tianjin-Hebei region (BTH2+26 cities) in December
2018, with average contributions up to 55%, 58%, 60%, and 57% for Handan, Shijiazhuang, Xingtai, and
Dezhou, respectively. Correspondingly, the inclusion of CPM emissions also narrowed the gap between
simulated and observed $PM_{2.5}$ concentrations over the BTH2+26 cities. These results improve the
simulation performance of atmospheric OA and $PM_{2.5}$, and may provide important implications for the
sources of OA.











## 1 Introduction

Atmospheric fine particulate matter (PM$_{2.5}$, particulate matter with aerodynamic diameter not exceeding 2.5 μm) is a serious and recurring air quality problem. Although the annual average concentration of PM$_{2.5}$ in China has declined in recent years, it still exceeds standards promulgated by the World Health Organization (WHO) Air Quality Guidelines (Lin et al., 2018). Heavy haze episodes occur frequently in winter, especially for the eastern regions in China (Li et al., 2015; Chen et al., 2019; Li et al., 2017a). Despite large reductions in primary emissions during the COVID-19 lockdown, several periods of heavy haze continued to occur in eastern China (Huang et al., 2021; Wang et al., 2020c, 2021). Organic aerosols (OA) contribute a large fraction to the PM$_{2.5}$ worldwide, ranging from 20% to 90% (Carlton et al., 2009; Kanakidou et al., 2005) with a negative impact on radiative climate forcing, air quality and human health (Gehring et al., 2013; Pope et al., 2002). POA comes from a variety of sources, including fossil fuels and biomass burning. SOA is generated through photochemical oxidation of volatile organic compounds (VOCs) followed by gas-particle partitioning of low-volatility organic compounds into the aerosol phase (Fuzzi et al., 2006; Kroll and Seinfeld, 2008) Currently, the significant contributions of OA to PM$_{2.5}$ and SOA to OA have been demonstrated in many observational results (He et al., 2020; Veld et al., 2021; Zhang et al., 2017). For example, Huang et al. (2015) explored the role of OA in PM$_{2.5}$ during a severe haze episode in Beijing, Shanghai, Xi'an and Guangzhou, showing the substantial contribution of OA to PM$_{2.5}$ (30~50%) and SOA accounted for 30~77% of OA. Sun et al. (2015) showed that OA constituted up to 65% of submicron aerosols during winter in Beijing, with 38% being SOA.

With respect to chemical schemes of SOA formations, a two-product model (Odum et al., 1996) was first proposed based on absorptive partitioning theory (Pankow, 1994) and chamber data. To address the underestimation of the early two-product model, the volatility basis set (VBS) framework was developed (Donahue et al., 2006). In this VBS scheme, semi-volatile and intermediate volatility precursors (S/IVOCs) were classified by their volatilities based on the absorptive partitioning theory (Robinson et al., 2007). A large portion of SVOCs are emitted as POA and then evaporate at ambient conditions due to gas-particle partitioning, while the IVOCs species exist in the form of organic vapor under many atmospheric conditions in the absence of photochemical reactions (Shrivastava et al., 2011). Currently, the VBS mechanism has been incorporated into many global and regional scale models (Lane et al., 2008; Murphy and Pandis, 2009; Shrivastava et al., 2008; Han et al., 2016). The two-dimensional (2-D) VBS scheme was put forward to improve the accuracy of fragmentation processes and OA oxidations (Donahue et al., 2011; Zhao et al., 2016). Despite advances in SOA formation mechanisms, a gap exists between observed



and modeled results due to uncertainties in parameterization of SOA yields, inapplicability of parameter
localizations caused by regional and sectoral differences and incomplete information on emission rates
and properties of SOA precursors. Recent studies have begun to focus on important effects of emissions,
including traditional precursors (VOCs) and low-volatility precursors (S/IVOCs). For example, Zhao et
al. (2017) found that IVOC of 1.5–30 times POA emissions contributed largely to OA concentrations over
the BTH region. Wu et al. (2019) constructed an inventory of S/IVOCs for the Pearl River Delta (PRD)
region in China and conducted a simulation using the WRF-Chem model leading to an increase of 161%
in SOA predictions. Emissions of S/IVOCs from mobile sources and IVOCs from volatile chemical
products were also parameterized in models to represent SOA formation (Jathar et al., 2017; Lu et al.,
2020; Pennington et al., 2021). Although the significant role of potential emission sources in OA
formation has been demonstrated, underestimation of SOA by current air quality models has not been
totally resolved. Stationary combustion sources are one of the major emission sources of $PM_{2.5}$, including
power plants and factories. Sampling temperatures and dilution rates are key factors for accurate
measurements of organic matter (Morino et al., 2018). The total primary PM emitted from stationary
sources is composed of filterable PM (FPM) and condensable PM (CPM). FPM exists in liquid or solid
phases, while CPM is in gas phase in flue (Corio and Sherwell, 2000; Feng et al., 2018). CPM is defined
by the U.S. Environmental Protection Agency (EPA, 2017) as particles which are gaseous at flue gas
temperature but condense or react in the ambient air to form solid or liquid PM through dilution and
cooling immediately after discharge. With ultralow emission standards implemented by coal-fired power
plants (<10 $mg/Nm^3$) since 2014, FPM emissions have been substantially reduced (even below 5 $mg/Nm^3$)
(Tang et al., 2019), making the remaining emissions of CPM an important issue. The Ministry of Science
and Technology of China issued a national key research and development project on the causes and
controls of air pollution in 2016, which mentioned key technologies for controlling CPM emissions
(http://www.acca21.org.cn/zdy_cms/siteResources/DisasterReduction/resources/otherfiles/
20160425/f15345793.pdf). The current measurement studies about emission characteristics and chemical
composition of CPM exhibited non-negligible emissions. For example, Yang et al. (2014, 2018a, 2018b)
conducted investigations for different types of industrial boilers and power plants, and concluded that
CPM constituted 25.7~96.5% of $PM_{2.5}$. For an ultralow-emission coal-fired power plant, Li et al. (2017b)
reported that the emission concentrations of CPM accounted for 83% of the $PM_{2.5}$. Wang et al. (2018)
calculated the average emission factors of CPM from two stacks in a waste incineration power plant to
be 0.201 and 0.178 g/kg, which were 22.0 and 31.2 times higher than the corresponding those of FPM,





respectively. Wu et al. (2020) found that FPM emissions from four typical coal-fired power plants met
Chinese ultra-low emission standards, while CPM showed high levels (even above 10 mg/Nm$^3$). CPM
includes organic and inorganic components, known as organic CPM and inorganic CPM, respectively.
The contributions of organic fractions varied from 13.6% to 80.5%, depending on different fuel types,
test methods and operating conditions (Lu et al., 2019; Song et al., 2020; Yang et al., 2021, 2018b). Many
studies confirmed that more than 50% of organic composition were measured in CPM (Li et al., 2017c,
2017d; Song et al., 2020; Wu et al., 2020), revealing that organic matter comprising a large proportion in
CPM needed to be taken into account. These above studies provided valuable basic information of CPM
emission characteristics for data references in this study, as summarized in Table S3. It is likely that the
inorganic fractions of CPM make a contribution to the water-soluble ions in PM$_{2.5}$, and organic
components contribute to the organic matter in PM$_{2.5}$. In addition, large amounts of low volatile organic
compounds in CPM can be important precursors for SOA formation.
Current measurement methods for PM in stationary exhaust sources in China (GB/T 16157-1996)
have not involved the collections of CPM; and the chemical composition of collected PM was quite
different from that actually released into the atmosphere (Hu et al., 2016). The emission inventory
constructed based on emission surveys did not include the CPM emissions. So it is important to introduce
CPM emissions to the current emission inventory. For example, a European study improved OA
simulations by including the CPM emissions from residential wood combustion sources (Van Der Gon et
al., 2015). Morino et al. (2018) revised the emission inventory by the consideration of CPM in Japan and
showed that the OA emission rates were up to seven times the previous ones and CPM contributed largely
to atmospheric OA concentrations. A shortcoming of that study was that it did not separate the effects of
CPM emissions on POA and SOA concentrations. Moreover, studies still lack quantification of emissions
of CPM released by stationary combustion sources in China.
In this study, we use the available CPM emission information to construct an emission inventory of
CPM from stationary combustion and mobile sources in China (with a focus on OA) and conducted 15
sensitivity simulations to explore the contributions of CPM emissions to atmospheric OA and PM$_{2.5}$
concentrations during the winter haze episodes over China. This quantitative study about organic CPM
emissions and the roles of CPM in the OA formation emphasizes the importance of constraining CPM
emissions from stationary combustion and mobile sources.

**2 Materials and methods**



**2.1 Estimations of CPM emissions**

We collected available emission measurement data of CPM based on published literatures. Totally, CPM emission data from 52 stationary combustion sources were acquired (Table S3). The emission sectors for these data included coal-fired power plants, waste incineration power plants, industrial coal boilers, heavy oil boilers, wood boilers, natural gas boilers, diesel boilers, iron and steel plants, and incinerators. Emissions of CPM depend on many factors including source categories, fuel types, sampling flue gas temperature, and air pollution control devices (Feng et al., 2021). Also, different measurement methods produced different results of CPM emissions (Wang et al., 2020a). Recently, cooling and dilution methods have been applied to monitor CPM concentrations. The emission rate of OA in CPM was estimated as follows in Eq. (1) and (2) (Morino et al., 2018):

$$E_{OA}(CPM) = \sum A \times EF_{OA}(CPM) = \sum A \times EF_{PM2.5}(FPM) \times \frac{EF_{OA}(CPM)}{EF_{PM2.5}(FPM)} \qquad (1)$$

$$E_{OA}(CPM) = \sum E_{PM2.5}(FPM) \times \frac{C_{OA}(CPM)}{C_{PM2.5}(FPM)} \qquad (2)$$

$$E_{OM_{lsi}}(CPM) = E_{OA}(CPM) \times \frac{E_{OM_{lsi}}(CPM)}{E_{OA}(CPM)} = E_{OA}(CPM) \times \frac{C_{OM_{lsi}}(CPM)}{C_{OA}(CPM)} \qquad (3)$$

Where $E_{OA}$(CPM) is the emission rate of organic matter in CPM; $EF_{OA}$(CPM) is the emission factor of organic matter in CPM; $E_{PM2.5}$(FPM) is the emission rate of $FPM_{2.5}$; $EF_{PM2.5}$(FPM) is the emission factor of $FPM_{2.5}$; $A$ denotes the activity level; $C_{OA}$(CPM) is the concentration of organic matter in CPM; and $C_{PM2.5}$(FPM) is the concentration of $FPM_{2.5}$. Among these parameters, $C_{OA}$(CPM) and $C_{PM2.5}$(FPM) were derived from the collected emission survey data at the above stationary sources. The ratios of $C_{OA}$(CPM) to $C_{OA}$(FPM) can be used to estimate $E_{OA}$(CPM), but due to the limited data and very low values of $C_{OA}$(FPM) at stationary sources, $C_{PM2.5}$(FPM) was used instead of $C_{OA}$(FPM). The ratios of $E_{OA}$(CPM) to $E_{PM2.5}$(FPM) and $EF_{OA}$(CPM) to $EF_{PM2.5}$(FPM) should be equal to the ratios of $C_{OA}$(CPM) to $C_{PM2.5}$(FPM) at the same dilution ratio in the emission surveys, as summarized in Table 1. In this estimate, these emission ratios collected from the best available data from the stationary sources were applied to represent the stationary combustion sources in the current emission inventory. $A$ and $EF_{PM2.5}$(FPM) in Eq. (1) were combined to get $E_{PM2.5}$(FPM) in Eq. (2), acquired from $PM_{2.5}$ emission rates in the emission inventory of baseline year. The organic CPM mainly contains alkanes (with $C_{10}$-$C_{30}$ being the major n-alkanes), esters, and polycyclic aromatic hydrocarbons (PAHs) (Li et al., 2017c, d; Song et al., 2020; Zheng et al., 2018). Based on the relationship between carbon number of n-alkanes and saturation concentrations (C*) following Lu et al. (2018), it is reasonable to speculate that organic CPM is composed of organic matter





which has low volatility (LVOC, $10^{-3}<C^*<10^0$), semi-volatile (SVOC, $10^0<C^*<10^3$), or has intermediate
volatility (IVOC, $10^3<C^*<10^6$), combined as $OM_{lsi}$ (CPM). Since the volatility characteristics of organic
CPM from stationary sources have not been accurately determined in relevant measurement studies, the
emissions of $OM_{lsi}$ (CPM) were scaled to emissions of OA (CPM) in this estimate as shown in Eq. (3).
$E_{OMlsi}$ (CPM) is the emission rate of $OM_{lsi}$ in CPM; $C_{OMlsi}$ (CPM) is the concentration of $OM_{lsi}$ in CPM.
The specific partition coefficients for different volatility bins in the model will be discussed in the
following Sect. 2.3. In addition to stationary sources, mobile sources also generate certain emissions of
CPM. Due to the lack of CPM emission data from on-road and off-road vehicles, we increased OA
emission rates of the transportation sector (TR) by 30% to consider the contributions of CPM from these
sources, following Morino et al. (2018) and Lu et al. (2020).

**2.2 The model configuration**

The three-dimensional Community Multiscale Air Quality (CMAQ, v5.3.2) model developed by the

U.S. Environmental Protection Agency was used to simulate spatiotemporal distributions of chemical
species. The detailed model configuration can be referred to Appel et al. (2021) and Yu et al. (2014). The
gas-phase chemical mechanism was based on the Carbon Bond Mechanism 6 (CB6) scheme. The aerosol
module was based on the seventh-generation aerosol module of CMAQ (AERO7). The CMAQv5.0.2-
VBS version with AERO6 coupled with a VBS module (AERO6VBS) was used for comparison.
Compared to the SOA formation in AERO6 in the CMAQv5.2, the AERO7 module includes
improvements: enhanced consistency of the SOA formation pathways between chemical mechanisms
based on CB and SAPRC, updated photooxidized monoterpene SOA yields (Xu et al., 2018), added
uptake of water by hydrophilic organics (Pye et al., 2017), consumption of inorganic sulfate when forming
isoprene epoxydiol organic sulfate (Pye et al., 2013), and replacement of the Odum two-product model
with a VBS framework to parameterize SOA formation (Qin et al., 2021; Appel et al., 2021). Both
AERO6VBS and AERO7 contained five classes of organic matter with one class being nonvolatile and
the other four classes being semi-volatile with effective saturation concentrations of 1, 10, 100, and 1000
$\mu g\ m^{-3}$. Each of these volatility bins was assigned to the CMAQ species of LVPO1, SVPO1, SVPO2,
SVPO3 and IVPO1, respectively. The emissions of unspeciated IVOC were set equal to 1.5 times the
POA emissions in AERO6VBS and 6.579 times in AERO7 by default (Table 3). The high scale factor of
6.579 in AERO7 was set to consider missing pathways for the SOA formation from combustion sources
including the IVOC oxidation (Murphy et al., 2017; Murphy et al., 2021), and it was primarily



parameterized in Los Angeles where vehicle emissions are a principal source (Hayes et al., 2015). This
parameter setting may not be suitable for fire and wood-burning sources and might have accounted for
the CPM contributions. In addition, the ratios of $C_{SIVOCs}$ to $C_{OA}$ can be dependent on $C_{OA}$ under stack
conditions, which were generally above 3000 μg m$^{-3}$ in CPM according to the emission surveys.
Considering high OA concentrations, we revised the scale factor of IVOC to 1.5 (same as that in
AERO6VBS) in the AERO7_adj case which was regarded as the base case. Meteorological fields were
predicted by the Weather Research and Forecasting (WRF) model version 3.7. The physical schemes of
WRF were the same as those in Wu et al. (2018) and Zhang et al. (2021). Meteorological initial and
boundary conditions were provided by the National Center for Environmental Prediction (NCEP) final
analysis dataset with the spatial resolution of 1°×1° and temporal resolution of 6 h. The first several days
were used for model spin-up, varied for different pollution periods as described in Sect. 2.4. The gridded
anthropogenic emission data for 2014 and 2017 were derived from Emission Inventory of Air Benefit and
Cost and Attainment Assessment System (EI-ABaCAS) developed by Tsinghua University (Dong et al.,
2020; Zheng et al., 2019). It contained primary species such as $PM_{2.5}$, $SO_2$, $NO_x$, CO, NMVOCs, $NH_3$,
BC, and OC from nine anthropogenic sectors (i.e., agriculture, power plant, industry process, industry
combustion, steel, cement, residential, transport, and open burning). Biogenic source emissions were
calculated by on-line Biogenic Emission Inventory System version 3.14 (BEISv3.14) model (Carlton and
Baker, 2011). Dust emissions were calculated by an on-line windblown dust scheme (Choi and Fernando,
2008). Our study period in 2014 occurred before and during the Asia-Pacific Economic Cooperation
(APEC) summit held in Beijing (November 5–11, 2014). During the period of pre-APEC (October 28–
November 2) and full-APEC (November 3–11), some pollution control measures were gradually
implemented in Beijing and its surrounding areas. Thus we conducted emission reduction by 30% during
the above time period for two municipalities (Beijing and Tianjin), four provinces (Hebei, Shanxi, Henan,
and Shandong), and Inner Mongolia Autonomous Region (Li et al., 2017e, 2019). The simulation domain
covered mainland China by a 395 × 345 grid with the horizontal grid resolution of 12 km (Fig. 1). There
were 29 vertical layers in σ$_z$ coordinate system reaching the upper pressure (100 hPa) with 20 layers
located in the lowest 3 km to resolve the planetary boundary layer.

**248    2.3 Design of sensitivity simulation cases**

According to the emission parameters summarized in Table 1, we carried out bootstrapping and
Monte Carlo simulations to obtain the mean and uncertainty ranges of $E_{OA}(CPM)/E_{PM2.5}(FPM)$ for





stationary combustion sources including power plant (PP), industry combustion (IN), steel (IR) (see Table
2). First, the optimal probabilistic distributions and uncertainty ranges were determined for each source
category. Then the statistical bootstrap simulation was applied to calculate the mean and 95% confidence
interval of emission ratios for each source category. Finally, the uncertainties of these parameters were
propagated to calculate the total uncertainty of emission by running Monte Carlo simulations for 10,000
times. On this basis, a series of sensitivity cases including low, medium, and high emission ratios were
designed to explore the contributions of CPM emissions to OA concentrations and quantify uncertainty
ranges of CPM effects on OA (Table 3).
Here, to explore the contributions of organic CPM emissions to atmospheric OA and $PM_{2.5}$
concentrations, the estimated emissions of organic CPM were added into the CMAQ model as an
individual source, separated from other emission sources. For the base scenarios, the simulations were
performed with the input of the previous emission inventory without the newly constructed organic CPM
emissions in the AERO6VBS, AERO7_def and AERO7_adj cases. Except the revision of scale factor of
IVOC in AERO7_adj case, the rests were kept at the default settings in the model. On the other hand,
different volatility distributions could be chosen for different emission sources, but this was not our study
focus and did not interfere with the results of CPM contributions. For the other cases including CPM
emissions from stationary combustion sources, the emissions of organic CPM were mapped to surrogate
species of different volatility bins (LVPO1, SVPO1, SVPO2, SVPO3, and IVPO1) in the CMAQ model.
Due to the unavailable volatility distribution information of $OM_{lsi}$ (CPM), different scaling factors of
volatility bins were employed under each emission scenario to discuss the uncertainty of CPM effects. In
this study, we tested two kinds of scaling factors for the five volatility bins of SVOC: fac1 (0.09, 0.09,
0.14, 0.18, 0.5) (Grieshop et al., 2009), fac2 (0.40, 0.26, 0.40, 0.51, 1.43) (Shrivastava et al., 2011). The
fac2 estimated total SVOC emissions as 3 times POA emissions to consider missing $OM_{lsi}$ (CPM)
emissions. Although the high coefficient settings may lead to overestimation of the simulations, it was
still applied to discuss the sensitivity of modeling results to different volatility distributions. Then the fac3
(0.245, 0.175, 0.27, 0.345, 0.965) which is the average of fac1 and fac2, was also tested for the SVOC
volatility bins under which the IVOC scale factor was set to 2.5. The fac1, fac2, and fac3 were applied to
the $OM_{lsi}$ (CPM) emissions for cases S1.1, S1.2, and S1.3, respectively (see Table 3). For an evaluation
of the sensitivity of OA outputs to organic CPM emissions, we conducted simulations with different
magnitudes of CPM emissions at the 95% and 50% confidence interval. Thus the S2-S3 cases were
designed with the uncertainty ranges of $E_{OA}$(CPM)/$E_{PM2.5}$(FPM) at 95% confidence interval (73% and



128% of the amounts in S1), and the S4-S5 cases with the uncertainty ranges at 50% confidence interval
(90% and 109% of the amounts in S1). Moreover, the contributions of individual emission categories
including PP, IN, IR, and TR were quantified by excluding perturbation of other sources in the S6-9 cases.
The simulated contributions of CPM emissions to POA, SOA, OA, and $PM_{2.5}$ concentrations were
calculated as the improved concentrations after including CPM emissions relative to the base case under
these scenarios.

**2.4 Observational data**
For the year 2014, the simulation period was from October 6 to November 14, 2014, with the first 8
days being the model spin-up time. Field observation data during the episode from October 14 to
November 14, 2014, at the Institute of Atmospheric Physics (IAP) (39°58′ N, 116°22′ E) in Beijing were
from Li et al. (2017a) and Xu et al. (2015). Concentrations of aerosol components were measured in $PM_1$.
In order to make a comparison between simulated and observed results, the $PM_1/PM_{2.5}$ ratio of 0.77 was
used to calculate the observed component concentrations in $PM_{2.5}$ based on the observations from Xu et
al. (2015). Observation data of organic carbon (OC) on November 3, 2014, at Qianyanzhou (located in
Jian city) and Changsha were provided by CERN Atmospheric Science Branch of the Institute of
Atmospheric Physics, Chinese Academy of Sciences (Liu et al., 2018). For the year 2018, the simulation
period included December 1 to 31, 2018, with the first 5 days for model spin-up. The observation values
of OC in the BTH2+26 cities were provided by China Environmental Monitoring Station. These cities
include Beijing, Tianjin, Anyang, Baoding, Binzhou, Cangzhou, Changzhi, Dezhou, Hebi, Handan,
Hengshui, Heze, Jincheng, Jinan, Jining, Jiaozuo, Kaifeng, Liaocheng, Langfang, Puyang, Shijiazhuang,
Tangshan, Taiyuan, Xingtai, Xinxiang, Yangquan, Zibo, and Zhengzhou. The OA/OC ratio of 1.4 (Simon
et al., 2011) was used to calculate OA concentrations for the comparison with the simulation results. The
observed concentrations of $PM_{2.5}$ were collected from the Chinese National Environmental Monitoring
Center (CNEMC). Since the $PM_{2.5}$ observation data from December 22 to 26 were missing, the following
analysis of $PM_{2.5}$ did not include these five days. The hourly observation data of meteorological factors,
including temperature (T), relative humidity (RH), wind speed (WS), and wind direction (WD), were
provided by the China Meteorological Administration (http://data.cma.cn/site/index.html).

**3 Results and discussion**
**3.1 Emissions of condensable particulate matter**





Emissions of OA in CPM ($E_{OA}$(CPM)) were comparable to or even exceeded the emissions of
filterable $PM_{2.5}$ ($E_{PM2.5}$(FPM)) for most stationary combustion sources, regardless of the differences
among these values (Table 1). Therefore, we constructed a new emission inventory by including CPM.
The annual emissions of OA in previous and modified emission inventory over China for the year 2014
and 2017 are presented in Fig. 2. The OA represents the organic matter in the emission input before the
further volatility distributions, while OM ($C^* \leq 100$) represents the organic matter allocated in the bin of
$C^*$ equal to 10 and below after application of the volatility distributions for the fac1, fac2 and fac3 cases.
Based on the simulation case settings, OA (FPM) from all the sectors was multiplied by fac1 (0.5), while
OA (CPM) from stationary combustion and mobile sources was multiplied by fac1 (0.5), fac2 (1.57) or
fac3 (1.035). In the previous inventory for 2014 without CPM, the emissions of OA were 3664.6 Gg,
approximately equal to 40% of $PM_{2.5}$ emissions. After the inclusion of CPM released by stationary
combustion sources in the new inventory, the emissions of OA were enhanced by a factor of 2 and even
exceeded emissions of $FPM_{2.5}$. The dominant contributors of OA (FCPM) are combustion sources in
power plant and industrial sectors, estimated to be 66% (7006.2 Gg) of the total OA emissions
(10531.1Gg). The emissions of OM ($C^* \leq 100$) remained unchangeable for the open burning, domestic,
and industry process sources since they were mostly FPM, while OM ($C^* \leq 100$) for the power plant,
industry combustion, and steel sources were variable based on whether fac1, fac2 or fac3 were applied
to the CPM. Similarly, the emissions of OA (FCPM) were 3 times those of OA (FPM) for the year 2017.
The emissions of OA from power plant, industry combustion, and steel sources increased by 33 times
after considering CPM emissions. These results indicate that the inclusion of organic CPM from
stationary combustion sources has a major impact on OA emissions and improves contributions of
industrial and power sectors to OA emissions.
Notably, the emission estimates of OA in CPM contained uncertainties, mainly attributed to the
representativeness and limitations of chosen emission sources. For power plant, industry combustion, and
steel sectors, the average ratios of $E_{OA}$(CPM) to $E_{PM2.5}$(FPM) were 4.12, 1.38 and 2.80, respectively (Table
2). Overall, the uncertainty range of $E_{OA}$(CPM) related to variabilities in the ratio of $E_{OA}$(CPM) to
$E_{PM2.5}$(FPM) was -27% ~ +28% at the 95% confidence interval. On this basis, a series of sensitivity cases
with different emission ratios were set to determine the uncertainty ranges of CPM contributions (Table
3). In the future, actual measurements of organic CPM emissions from various sources and source-specific
identification of volatility distributions are needed to reduce uncertainties in emission estimates.



**3.2 Meteorological evaluation**

Comparisons between simulated and observed hourly meteorological variables including T, RH, WS, and WD from October 14 to November 14, 2014, at the Beijing site are displayed in Fig. S1. Results show that the model reproduced the hourly variations of T and RH reasonably well, although the maximum and minimum T, and RH did not totally match the observed values. The simulated WS were overestimated, but the hourly changes were reproduced. The variations of WD were not well captured, but the magnitudes of simulated WD were consistent with the observations over the whole period. A more detailed model evaluation for meteorological variables during October 14 –November 14, 2014 and December 1–30, 2018 at 9 cities over China is given in Table S1. MB, GE, RMSE denote the bias, root mean square error, and fractional error, respectively, and R refers to the correlation coefficient between observed and simulated results. For the Beijing site in 2014, the MB of T was -0.3 □, indicating a small deviation of modeled temperature. Good correlations between simulation and observation were shown for T, RH, and WS with R values of 0.90, 0.75, and 0.62, respectively. For all these cities, T, RH, and WS had the R values of 0.83~0.94, 0.67~0.89, and 0.21~0.70 during the study period in 2014, respectively. The R values for T, RH, and WS in 2018 were 0.74~0.95, 0.52~0.85, and 0.33~0.75, respectively. The GE and RMSE of WS were lower than model performance criteria (2 m/s) (Emery et al., 2001) for most cities, displaying relatively good simulations of wind speed. In summary, the WRF model showed a relatively consistent simulation performances of meteorological variables.

**3.3 Effects of CPM emissions on POA and SOA concentrations**

For the hourly observed and simulated SOA and POA concentrations at the Beijing site, Figs. 3 and 4 show obvious improvements of SOA and POA levels after the consideration of CPM contributions. The specific model species for POA and SOA are shown in Table S4. In all the simulation scenarios, five complete ascending and descending SOA episodes in Fig. 3 were well captured, with much lower mean bias between observations and simulations than previous results of Li et al. (2017a). Three pollution episode processes before the APEC were clearly captured by the model. The third process (October 27– November 1) had lower observed SOA levels relative to the first (October 16–21) and second processes (October 22–26), attributed to lower precursor emission concentrations, lower temperature, and regional transports by strong northerly winds on October 26. During the APEC, there were two pollution episodes with lower SOA concentrations due to the effects of emission controls and meteorological conditions (Ansari et al., 2019; Liang et al., 2017). Compared to the observed values, cases without CPM exhibited



varying degrees of overestimation or underestimation for SOA and POA. For example, in the AERO7_def case, the maximum SOA values were overestimated by 42% in the first episodes and up to 67% in the third episodes, while the POA values were largely underestimated by an average of 73% during the whole time period. Then we revised the scale factor of IVOC in the AERO7_adj case (see Table 3). The overestimated SOA in the AERO7_def for the first and third episodes were reduced by 65% and 60%, respectively. In comparison, the AERO6VBS case underpredicted SOA by up to 65%, and simulated low levels of POA during the first three periods and high levels in the last two episodes. The base case in the following discussions referred to AERO7_adj. Overall, the base case underestimated the average POA and SOA levels by 76% and 66% (Table 4), respectively, emphasizing the potential contributions of missing CPM sources.

After considering CPM emissions, the underestimation of average POA and SOA was reduced to 38% and 24% under the S1.1 scenario, respectively (Table 4). From the simulated hourly variations in the S1.1 case (Fig. 3), SOA concentrations were enhanced by factors of 0.01~3.10 relative to base case, more consistent with the observations. The gap between average simulations and observations decreased from -11.56 to -4.23 $\mu g\ m^{-3}$ (63% decrease). For the peak values in the first, second, fourth, and fifth pollution episodes, the improvements in the peak SOA concentrations were approximately 30, 30, 10, and 15 $\mu g\ m^{-3}$. Nevertheless, the overestimation of SOA occurred in the third process, mainly due to meteorological conditions considering the fact that the observed and modeled wind directions were inconsistent during this period as shown in Fig. S1. The prevailing southerly and northeast wind directions in the model during the third process did not bring clean air from the northwest boundary to dilute the local generated SOA (Li et al., 2016, 2019). Also, higher simulated wind speeds transported more precursors with the southerly and northeast winds and caused the overestimation of SOA (see Fig. S1). Correspondingly, the hourly POA simulation concentrations in the S1.1 case increased by 0.13~4.55 times compared to the base case, narrowing the average gap between simulations and observations from -12.29 to -6.14 $\mu g\ m^{-3}$ (50% decrease), but the high observed levels of POA were still not attained under this scenario. Comparatively, the S1.2 case presented similar hourly simulation results of SOA to the S1.1 case with the enhancement by factors of 0.02~3.77 versus the base case, while the simulated POA values were nearly 1.3 times higher than the S1.1 case, capturing most of the high observations throughout the whole study period. This demonstrates that the SVOC parameters had more impact on POA than SOA. Under the S1.3 scenario using different SVOC and IVOC parameters from the S1.2 case, the simulation concentrations of SOA were 24% higher and POA were 29% lower than those under the S1.2 scenario as shown in Table





4. Based on the evaluation results, the S1.3 scenario showed the optimal improvement effects, with the
mean biases of 1.16% for POA and 2.16% for SOA (see Table 4). In consideration of the uncertainty
ranges of CPM emissions, a series of sensitivity cases with different emission ratios were conducted.
Under the minimum emission scenario in the S2.1 case, the average SOA and POA concentrations were
14.4%, and 16.5% lower than those in the S1.1 case, respectively. Under the maximum emission scenario
in the S3.1 case, the average SOA and POA concentrations were 14.6% and 17.3% higher than those in
the S1.1 case, respectively. Thus the model can resolve 62% (52%~73%) of the observed POA
concentrations and 76% (65%~87%) of the observed SOA concentrations in the cases S1.1 (S2.1, S3.1).
Then the S2.2 and S3.2 cases applied the same S/IVOC parameters as S1.2, and also displayed similar
results of SOA to those in the S2.1 and S3.1 cases, respectively. Under this setting, the uncertainty ranges
were -14.6% to +14.5% for SOA, and -22.8% to +23.9% for POA in the S1.2 case as shown in Table 4.
For the S4.2 and S5.2 cases with the CPM emissions at 50% confidence interval, their SOA concentrations
showed small changes with 5.3% lower in the S4.2 case and 4.7% higher in the S5.2 case than the S1.2
case; similar minor sensitivity of 8.5% decrease (S4.2) and 7.6% increase (S5.2) were found for POA. To
explore the contribution of each source category to SOA and POA and identify the key anthropogenic
sources of CPM, we conducted simulations with the different separate inputs (S6~S9) (see Table 3).
Results show that the CPM emissions from the IR sector made the largest contribution to the POA and
SOA increases, accounting for 59% of POA and 55% of SOA, followed by PP (26% for POA and 30%
for SOA) and IN sources (13% for POA and 14% for SOA). This was consistent with the differences in
the CPM emissions from the above three source sectors (Fig. 2). The sensitivities of SOA and POA to the
emission ratio of organic CPM from the TR sector were very small, indicating a weak impact on OA due
to small contributions of transportation sources to the OA emissions in FCPM. The above results
demonstrate that CPM from stationary sources was an important source for both POA and SOA formations.
In summary, when considering the uncertainties of organic CPM emissions, CPM can be a significant
contributor to OA concentrations, with the contributions of 61% (53%, 67%) to POA, 55% (48%, 61%)
to SOA, 58% (50%, 63%) to OA under the S1.1 (S2.1, S3.1) scenario, and 83% (78%, 86%) to POA, 59%
(52%, 64%) to SOA, 74% (67%, 78%) to OA under the S1.2 (S2.2, S3.2) scenario. The S1.3 scenario had
the best improvement performance with CPM contributing 76% to POA, 67% to SOA, and 71% to OA.
Because of the better representations of temporal variations of SOA and POA after including CPM
emissions, OA simulations were correspondingly improved. To separate the effects of CPM on OA into
different process contributions, we compared simulation results of these sensitivity cases as shown in Fig.





5. The OA composition contains POA, ASOA (SOA from anthropogenic VOCs), BSOA (SOA from
biogenic VOCs), and SISOA (SOA from low volatile S/IVOCs). The difference between simulations and
observations decreased from 23.84 µg m$^{-3}$ in the base case to 10.37 µg m$^{-3}$ in the S1.1 case (56% decrease),
with the uncertainty of 13.95 µg m$^{-3}$ (41% decrease in S2.1) to 6.69 µg m$^{-3}$ (72% decrease in S3.1) relative
to the base case. However, these cases still underestimated the observed OA levels. The S1.2, S2.2 and
S3.2 cases increased the contributions of CPM to OA by 14.01, 10.24, 17.92 µg m$^{-3}$, with the percentage
increases of 60%, 52%, 66% compared to S1.1, S2.1 and S3.1, respectively. Notably, the average OA
simulations were the closest to the observations in S1.3, with the average CPM contributions of 24.41 µg
m$^{-3}$ and a minor overestimation of 1.68% (see Table 4). Taking OA composition into account, POA and
SISOA accounted for the largest part in all these scenarios. The effects of CPM were only reflected in the
enhancements of POA and SISOA. These results suggest that OA was sensitive to the emissions of CPM
and S/IVOCs, so it is required to reduce emission uncertainties for better simulations. To sum up, the
revised simulations after the inclusion of CPM from stationary combustion and mobile sources led to
improved modeling performances of OA during the winter haze episodes, revealing a significant
contribution of CPM to atmospheric OA.

**3.4 Effects of CPM on OA and PM$_{2.5}$ concentrations**
To ensure the accuracy and reliability of our modeling results, further studies in other cities were
presented. Fig. 6 shows large contributions of CPM to OA on November 3, 2014, at Changsha and
Qianyanzhou. After the inclusion of CPM effects in the S1.1, S1.2 and S1.3 cases versus the base case,
the simulated OA concentrations were improved by 96.4%, 198.3% and 142.1% for Changsha,
respectively. The simulated OA concentrations increased by 129.7%, 243.1% and 199.1% in the S1.1,
S1.2 and S1.3 cases versus the base case for Qianyanzhou, respectively. Comparatively, the S1.2 case
contributed to greater increases of OA concentrations, narrowing the simulation-observation bias from
80% to less than 40% for Changsha and more than 70% to less than 25% for Qianyanzhou. The remaining
bias was probably attributed to the effects of meteorological factors.
The impacts of CPM on OA were studied during December 6–30, 2018, in the BTH 2+26 cities.
Likewise, the improvements in daily OA simulation concentrations can be found at the four studied cities
after the consideration of CPM, especially for high pollution days (Fig. 7). The modeled underestimations
of OA were improved from -68% to -28%, -63% to -13%, -75% to -36%, and -71% to -33% with the
inclusion of CPM emissions in the S1.1 case relative to the base case for Handan, Shijiazhuang, Xingtai





and Dezhou, respectively (Table 4). The contributions of CPM emissions to total OA concentrations
reached up to 55%, 58%, 60%, and 57% for Handan, Shijiazhuang, Xingtai, and Dezhou, respectively.
Under the S1.3 scenario, the OA simulations showed greater increases, and slightly exceeded observation
values with the mean biases of 10%, 40%, 2%, and 3% for the above four cities, respectively. For example,
daily OA levels in Handan increased by 5.7~59.3 $\mu g\ m^{-3}$ after including CPM effects (S1.1 versus base
case). On average, CPM contributed to the increases in OA concentrations by 1.2 times. However, some
observations were not captured, while the observed value on December 20 was overestimated, indicating
uncertainties of the estimated organic CPM emissions. Under the S1.3 scenario, the average simulated
OA concentrations were enhanced by 2.4 times relative to the base case, with a good capture of some
underestimated values in the S1.1 case. For Shijiazhuang with daily OA concentrations below 80 $\mu g\ m^{-3}$,
the base case underestimated OA levels by 10~84%. After incorporating the CPM emissions in the S1.1
case, the daily OA concentrations were significantly improved by factors of 0.8~2.0. Some observed high
values of OA were well captured in the S1.1 case on December 10 with the simulation of 64.6 $\mu g\ m^{-3}$
versus observation of 58.6 $\mu g\ m^{-3}$, and on December 14 and 30. Under the S1.3 scenario, the daily OA
levels increased by factors of 1.7~4.4 relative to the base case. Although the average OA concentrations
were somewhat overestimated in the S1.3 case, good agreements between observations and simulations
existed on some days, including December 9, 12, 13, 16-19, and 24. For Xingtai, the simulated OA
concentrations were enhanced by factors of 1.2~2.4 in the S1.1 case relative to the base case. The model
can resolve 64% of average OA observations in the S1.1 case when the emissions of CPM were included.
The average OA simulation value was improved by 32.5 $\mu g\ m^{-3}$ in the S1.3 case compared to the base
case. Then Dezhou showed similar results with the enhancement of 1.0~2.2 times for daily OA contributed
by CPM in S1.1. Although the observed high OA concentrations exceeding 80 $\mu g\ m^{-3}$ on December 11
and 16 were not captured in the S1.1 case, the bias between simulations and observations was reduced to
-27.2 and -31.2 $\mu g\ m^{-3}$ versus -65.1 and -58.6 $\mu g\ m^{-3}$ in the base case, respectively. The underestimations
of high OA levels on December 11 and 16 were resolved in the S1.3 case, and the average concentrations
over the whole period were very close to the observations. Table S2 shows the model evaluation results
for PM$_{2.5}$ concentrations under different sensitivity simulation cases. Dezhou was not included due to the
missing data. After including the CPM emissions in the S1.1 case, the model can resolve 83%, 83%, and
69% of average PM$_{2.5}$ observations with increases in PM$_{2.5}$ concentrations by 35%, 40%, and 41% relative
to the base case for Handan, Shijiazhuang, and Xingtai, respectively. PM$_{2.5}$ simulations were further
enhanced for these four cities in the S1.3 case with the NMB values of 3%, 8%, and -11%, respectively.





It was notable that the emissions of inorganic components in CPM were not investigated in this study,
which can cause modeling deviation. Other factors including boundary layer height and wind can also
affect the simulations. In summary, our estimated CPM emissions showed a reasonable range, which can
make a significant contribution to atmospheric OA and $PM_{2.5}$.

**3.5 Regional contributions of CPM to OA and $PM_{2.5}$**

The regional effects of CPM emissions on atmospheric OA and $PM_{2.5}$ from a nationwide perspective

were investigated. The concentrations of POA, SOA and OA averaged over the whole study period from
October 14 to November 14, 2014, showed varying degrees of regional increases after incorporating CPM
emissions, mainly in central and eastern regions in China (Fig. 8). In the base case, the simulation values
of POA and SOA were both lower than 14 µg m$^{-3}$ over China. Correspondingly, OA concentrations did
not exceed 22 µg m$^{-3}$ with the maximum values distributed in the BTH region and Central China. After
the consideration of CPM effects in the S1.1 case relative to the base case, the concentrations of POA,
SOA and OA substantially increased over North China, East China, and Central China including Beijing,
Tianjin, Shanghai, and provinces of Liaoning, Shandong, Shanxi, Henan, Hubei, Anhui, Jiangsu, Zhejiang,
Hunan, Jiangxi. The most remarkable enhancement values were up to 10, 12, and 20 µg m$^{-3}$ for POA,
SOA and OA, respectively. Then under the S1.2 scenario with the same emissions as the S1.1 case but
different SVOCs parameterization, substantial increases in the POA simulations by more than 16 µg m$^{-3}$
were found for most cities in North China, East China, and Central China, with the maximum distributed
in the BTH region (up to 24 µg m$^{-3}$), attributable to large amounts of emissions from industrial plants and
power plants in this region. The OA concentrations for many cities located in North China and East China
increased by more than 24 µg m$^{-3}$ after including CPM emissions in the S1.2 case. Since the contributions
of CPM to SOA in the S1.2 case were only slightly larger than those in the S1.1 case, the greater
improvements of OA in S1.2 mainly result from the POA increases. The S1.3 case used different S/IVOCs
parameterizations from the S1.2 case, with the regional contributions of CPM emissions to POA and SOA
lower and higher than those in S1.2, respectively. The regional increases in the POA, SOA and OA
simulations in the S1.3 case were not lower than 10, 12, and 20 µg m$^{-3}$ for most cities in North China,
East China, and Central China, respectively.

The regional contributions of CPM emissions to $PM_{2.5}$ concentrations were explored in the

BTH2+26 cities averaged over the period from December 6 to 30, 2018 (Fig. 9). In the base case without
the CPM effects, the model comparisons against observations suggest that $PM_{2.5}$ levels were greatly





underestimated in almost all cities except Tangshan (Fig. 9a). Several cities with observed PM$_{2.5}$ concentrations higher than 80 µg m$^{-3}$ showed the greatest underestimations with simulation values under 50 µg m$^{-3}$. Under the S1.1 scenario including CPM emissions, the simulated PM$_{2.5}$ concentrations were substantially enhanced in almost all the studied cities, closer to the observations (Fig. 9b). The contributions of CPM to PM$_{2.5}$ were not lower than 14 µg m$^{-3}$ for the most cities (Fig. 9c). Under the S1.3 scenario, CPM made a significant contribution to PM$_{2.5}$ concentrations, more than 26 µg m$^{-3}$ for most cities (Fig. 9f). High observations for Baoding, Shijiazhuang, Xingtai, Hengshui, Dezhou and Handan were well captured (Fig. 9e). The scatter plots of observed and simulated daily PM$_{2.5}$ concentrations for all BTH2+26 cities in Fig. 9d show obvious improvement in PM$_{2.5}$ simulations after including CPM emissions, with the NMB values from -36.5% in the base case to -14.1% in the S1.1 case, and then to 6.8% in the S1.3 case. Nevertheless, there were still model-measurement biases for PM$_{2.5}$ concentrations in some cities with high observations exceeding 90 µg m$^{-3}$, including Baoding, Anyang, Puyang, Heze, Zhengzhou and Kaifeng. The insufficient improvement of PM$_{2.5}$ can be attributed to incomplete emission information of inorganic components, which need further research. In addition, some heavy pollution hours were chosen to investigate the regional impacts of CPM on PM$_{2.5}$ concentrations, including 8:00, 9:00, 10:00, 11:00, and 21:00 on December 15 (Fig. 10a). Besides the BTH2+26 cities, some surrounding cities (Chaoyang, Chengde, Datong, Dongying, Huludao, Jinzhou, Linxi, Luoyang, Luohe, Qinhuangdao, Qindao, Rizhao, Sanmenxia, Shangqiu, Shuozhou, Taian, Weihai, Weifang, Xinzhou, Xinyang, Yantai, Zaozhuang, Zhangjiakou, Zhoukou, Zhunmadian) were also included. Results show that the underestimated PM$_{2.5}$ concentrations in the base case were substantially improved after considering CPM emissions in S1.1 and S1.3, especially for some high observations over 170 µg m$^{-3}$. Better agreement between simulated and observed PM$_{2.5}$ concentrations for all these cities was achieved, with the NMB values from -36.0% in the base case to -15.3% in S1.1, and to 3.3% in S1.3 (Fig. 10b). To sum up, the consideration of CPM effects can improve the underestimation of regional OA and PM$_{2.5}$ simulations to a certain extent, especially during the heavy pollution periods.

## 4 Conclusions

In this study, we focused on emissions of condensable PM from stationary combustion and mobile sources and developed an emission inventory of organic CPM in China. Using emission inputs with and without CPM contributions, the CMAQ model was applied to simulate the impacts of CPM on atmospheric OA and PM$_{2.5}$ in China. The results show that the inclusion of CPM emissions increased





annual OA emissions by a factor of 2 for both the years 2014 and 2017. The power plant, industry
combustion, and steel sectors in the stationary combustion sources dominated OA emissions in the new
inventory. A series of sensitivity scenarios with different emission ratios and volatility distributions show
that CPM contributed significantly to the improvement of hourly SOA and POA concentrations during
the period from October 14 to November 14, 2014, at Beijing. The contributions of CPM were 53 ~ 86%
to POA and 48 ~ 67% to SOA under these scenarios. The model comparison against observations suggests
that the consideration of CPM effects improved the underestimations of simulation results and achieved
a good capture of peak SOA and POA values. In addition, the enhancements of daily OA levels by CPM
were demonstrated during December 6-30, 2018 at Handan, Shijiazhuang, Xingtai and Dezhou.
Compared to daily observations, the NMB values in these four cities were improved from -68%, -63%, -
75%, -71% (the base case) to -28%, -13%, -36%, -33% (the S1.1 case) for OA, respectively. The regional
contributions of CPM also narrowed the gap between simulated and observed concentrations of $PM_{2.5}$ in
the BTH2+26 cities. In conclusion, our estimated CPM emissions contributed significantly to the
improvements of simulation performances for both atmospheric OA and $PM_{2.5}$, especially during the high
pollution episodes. Therefore, the CPM emissions can be incorporated into chemical transport models
together with FPM to improve the simulation accuracies of OA and $PM_{2.5}$.
Our estimates of organic CPM emissions and SOA formation from CPM contained the following
uncertainties: (1) The construction of the organic CPM emission inventory in the present study was based
on the ratios of $E_{POA}$(CPM) to $E_{PM2.5}$(FPM) derived from limited sources, instead of the actual
measurement data of CPM emissions from the different sources and regions over China. (2) Since there
was no explicit volatility characterization of primary organic CPM species available for incorporation
into the emission inventories, the S/IVOCs emissions were scaled to the POA emissions. (3) Due to the
lack of relevant data, the original surrogate species of S/IVOCs and their properties in the CMAQ model
remained unchanged for representing the SOA formation from CPM, rather than introducing new model
species with identified parameters related to OH reaction rates, effective saturation concentration, and
multigenerational aging products. Based on these limitations, it is strongly recommended that future
studies conduct extensive surveys of CPM emissions from various stationary combustion sources and
measure the actual emissions of source-specific and region-specific S/IVOCs to better constrain OA
simulations by chemical transport models.

***Data availability.*** The emission data and model results are available upon request.



*Supplement.* The supplement related to this article is available online.

*Author contributions.* S.Y., P.L. conceived and designed the research. M. L. performed model simulations. M. L., X. C., Y. Z., and Z. L. conducted data analysis. Z. S., W. L., X. Z, B. N. M., K. A., R. M., D. R., and J. H. S contributed to the scientific discussions. M. Z, Y. S., Z. L., and C. S. provided observation data. S. W. provided the Abacas emission data. S. Y., M. L, P. L., and J. H. S wrote and revised the manuscript.

*Competing interests.* The authors declare that they have no conflict of interest.

*Disclaimer.* The views expressed in this article are those of the authors and do not necessarily represent the views or policies of the U.S. Environmental Protection Agency.

*Acknowledgements.* The authors would like to thank Comprehensive data collection and sharing platform for atmospheric environmental science (https://napcdata.craes.cn), and CERN Atmospheric Science Branch of the Institute of Atmospheric Physics, Chinese Academy of Sciences for providing OC measurement data.

*Financial support.* This study is supported by the National Natural Science Foundation of China (No. 42175084, 21577126, 41561144004, and 92044302), Department of Science and Technology of China (No. 2018YFC0213506 and 2018YFC0213503), and National Research Program for Key Issues in Air Pollution Control in China (No. DQGG0107). Pengfei Li is supported by National Natural Science Foundation of China (No. 22006030), Initiation Fund for Introducing Talents of Hebei Agricultural University (412201904), and Hebei Youth Top Fund (BJ2020032).

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





Table 1 List of the ratios of the emission rates of OA in condensable particulate matter (CPM)
($E_{OA}(CPM)$) to those of $PM_{2.5}$ in filterable particulate matter (FPM) ($E_{PM2.5}(FPM)$) from stationary
combustion sources based on the collected references.

| method | emission sources | number | $E_{OA}(CPM)/ E_{PM2.5}(FPM)$ | | | references |
|---|---|---|---|---|---|---|
| | | | [Min, Max] | Mean ± SD | median | |
| cooling method (EPA 202) | coal-fired power plant | 30 | [0.01, 25.4] | 6.87 ± 7.25 | 3.99 | Li et al. (2017c, 2017d); Li (2018); Li et al. (2019); Lu et al. (2019); Pei (2015); Qi et al. (2017); Song et al. (2020); Wang et al. (2020b); Wu et al. (2020); Yang et al. (2014, 2018b); Yang et al.(2021); Zhou (2019) |
| | waste incineration power plant | 2 | [1.64, 4.95] | 3.29 ± 1.65 | 3.29 | Wang et al. (2018) |
| | industrial coal-fired boiler | 6 | [0.14, 1.03] | 0.58 ± 0.34 | 0.50 | Lu et al. (2019) Yang et al. (2014, 2018a, 2018b) |
| | heavy oil-fired boiler | 4 | [0.28, 2.49] | 1.62 ± 0.88 | 1.85 | Yang et al. (2018a, 2018b) |
| | wood-fired boiler | 1 | | 0.03 | | |
| | natural gas-fired boiler | 1 | | 6.67 | | Yang et al. (2018a) |
| | diesel-fired boiler | 1 | | 15.84 | | |
| | iron and steel plants | 5 | [0.32, 7.22] | 3.35 ± 2.21 | 3.00 | Yang et al. (2014, 2015) |
| | incinerator | 1 | | 0.12 | | Yang et al. (2014) |
| dilution method (ISO 25597) | iron and steel coking plant | 1 | | 0.416 | | Zhang et al. (2020) |






Table 2 Probabilistic distributions with uncertainty ranges in the ratio of $E_{POA}$(CPM) to $E_{PM2.5}$(FPM) (95%
confidence interval). Para1 represents the mean for normal, and the mean of ln(x) for lognormal. Para2
represents the standard deviation for normal, and the standard deviation of ln(x) for lognormal. Mean
represents the mean for emission ratios of each source category derived from the statistical bootstrap
simulation.

| Input parameters | Emission sources | Distribution type | Para1 | Para2 | Mean | Uncertainty ranges (95% confidence level) |
|---|---|---|---|---|---|---|
| $E_{POA}$(CPM) /$E_{PM2.5}$(FPM) | Power plant | lognormal | 1.07 | 0.93 | 4.12 | (3.10, 5.29) |
| | Industry combustion | lognormal | -0.47 | 1.43 | 1.38 | (0.62, 2.44) |
| | Steel | normal | 2.80 | 1.98 | 2.80 | (0.92, 4.50) |
| Total | | | | | | (-27%, 28%) |






Table 3 Simulation case design. PP, IN, IR, and TR denote source sectors of power plant, industry combustion, steel, and transportation, respectively. Three kinds of scaling factors for the five volatility bins of organic CPM are tested: fac1 (0.09, 0.09, 0.14, 0.18, 0.5) (Grieshop et al., 2009), fac2 (0.40, 0.26, 0.40, 0.51, 1.43) (Shrivastava et al., 2011), and fac3 (0.245, 0.175, 0.27, 0.345, 0.965) which is the average of fac1 and fac2.

| Simulation Cases | Aerosol module | $E_{PP\_POA}(CPM)$ /$E_{PM2.5}(FPM)$ | $E_{IN\_POA}(CPM)$ /$E_{PM2.5}(FPM)$ | $E_{IR\_POA}(CPM)$ /$E_{PM2.5}(FPM)$ | Volatility bins |
|---|---|---|---|---|---|
| Only FPM | AERO6VBS | 0 | 0 | 0 | |
| | AERO7_def | 0 | 0 | 0 | |
| | AERO7_adj | 0 | 0 | 0 | |
| S1.1 | AERO7 | 4.12 | 1.38 | 2.80 | fac1 |
| S1.2 | AERO7 | 4.12 | 1.38 | 2.80 | fac2 |
| S1.3 | AERO7 | 4.12 | 1.38 | 2.80 | fac3 |
| S2.1 | AERO7 | 3.01 | 1.01 | 2.04 | fac1 |
| S2.2 | AERO7 | 3.01 | 1.01 | 2.04 | fac2 |
| S3.1 | AERO7 | 5.27 | 1.77 | 3.58 | fac1 |
| S3.2 | AERO7 | 5.27 | 1.77 | 3.58 | fac2 |
| S4.2 | AERO7 | 3.71 | 1.24 | 2.52 | fac2 |
| S5.2 | AERO7 | 4.49 | 1.50 | 3.05 | fac2 |
| S6_TR | AERO7 | 0 | 0 | 0 | fac1 |
| S7_IN | AERO7 | 0 | 1.38 | 0 | fac1 |
| S8_IR | AERO7 | 0 | 0 | 2.80 | fac1 |
| S9_PP | AERO7 | 4.12 | 0 | 0 | fac1 |




Table 4 Model evaluation statistics for hourly OA, POA and SOA concentrations during October 14–
November 14, 2014, and daily OA concentrations during December 6–30, 2018, under different
sensitivity simulation cases.

| Period | City | Species | Cases | N | OBS | SIM | MB | NMB | NME | R |
|---|---|---|---|---|---|---|---|---|---|---|
| October 14– November 14, 2014 | Beijing | OA | def | | 33.71 | 20.92 | -12.79 | -37.94% | 50.06% | 0.70 |
| | | | adj | | 33.71 | 9.87 | -23.84 | -70.73% | 70.80% | 0.70 |
| | | | S1.1 | 723 | 33.71 | 23.34 | -10.37 | -30.76% | 48.03% | 0.69 |
| | | | S1.2 | | 33.71 | 37.34 | 3.63 | 10.77% | 56.28% | 0.69 |
| | | | S1.3 | | 33.71 | 34.28 | 0.57 | 1.68% | 53.45% | 0.69 |
| | | POA | def | | 16.25 | 4.41 | -11.84 | -72.83% | 72.94% | 0.54 |
| | | | adj | | 16.25 | 3.96 | -12.29 | -75.61% | 75.66% | 0.54 |
| | | | S1.1 | 723 | 16.25 | 10.11 | -6.14 | -37.82% | 54.14% | 0.54 |
| | | | S1.2 | | 16.25 | 23.01 | 6.76 | 41.59% | 85.95% | 0.53 |
| | | | S1.3 | | 16.25 | 16.44 | 0.19 | 1.16% | 60.87% | 0.54 |
| | | SOA | def | | 17.46 | 16.50 | -0.96 | -5.47% | 50.42% | 0.73 |
| | | | adj | | 17.46 | 5.90 | -11.56 | -66.19% | 66.28% | 0.73 |
| | | | S1.1 | 723 | 17.46 | 13.23 | -4.23 | -24.20% | 47.47% | 0.72 |
| | | | S1.2 | | 17.46 | 14.33 | -3.13 | -17.90% | 47.61% | 0.72 |
| | | | S1.3 | | 17.46 | 17.84 | 0.38 | 2.16% | 53.38% | 0.72 |
| December 6–30, 2018 | Handan | OA | adj | | 45.24 | 14.66 | -30.58 | -67.58% | 67.58% | 0.62 |
| | | | S1.1 | 25 | 45.24 | 32.37 | -12.87 | -28.45% | 39.29% | 0.60 |
| | | | S1.3 | | 45.24 | 49.69 | 4.45 | 9.84% | 40.03% | 0.59 |
| | Shijiazhuang | OA | adj | | 42.22 | 15.57 | -26.65 | -63.12% | 63.12% | 0.61 |
| | | | S1.1 | 25 | 42.22 | 36.70 | -5.52 | -13.07% | 36.02% | 0.61 |
| | | | S1.3 | | 42.22 | 59.07 | 16.85 | 39.90% | 48.84% | 0.61 |
| | Xingtai | OA | adj | | 42.22 | 10.64 | -31.58 | -74.80% | 74.80% | 0.56 |
| | | | S1.1 | 25 | 42.22 | 26.89 | -15.33 | -36.31% | 44.14% | 0.57 |
| | | | S1.3 | | 42.22 | 43.12 | 0.90 | 2.13% | 34.85% | 0.57 |
| | Dezhou | OA | adj | | 41.66 | 12.07 | -29.59 | -71.02% | 71.02% | 0.48 |
| | | | S1.1 | 23 | 41.66 | 28.10 | -13.56 | -32.55% | 41.91% | 0.55 |
| | | | S1.3 | | 41.66 | 42.98 | 1.32 | 3.17% | 43.63% | 0.57 |

Note: OBS and SIM denote mean concentrations (μg m$^{-3}$) of observations and simulations, respectively; MB: mean bias;
NMB: normalized mean bias; NME: normalized mean error; R: correlation coefficient.



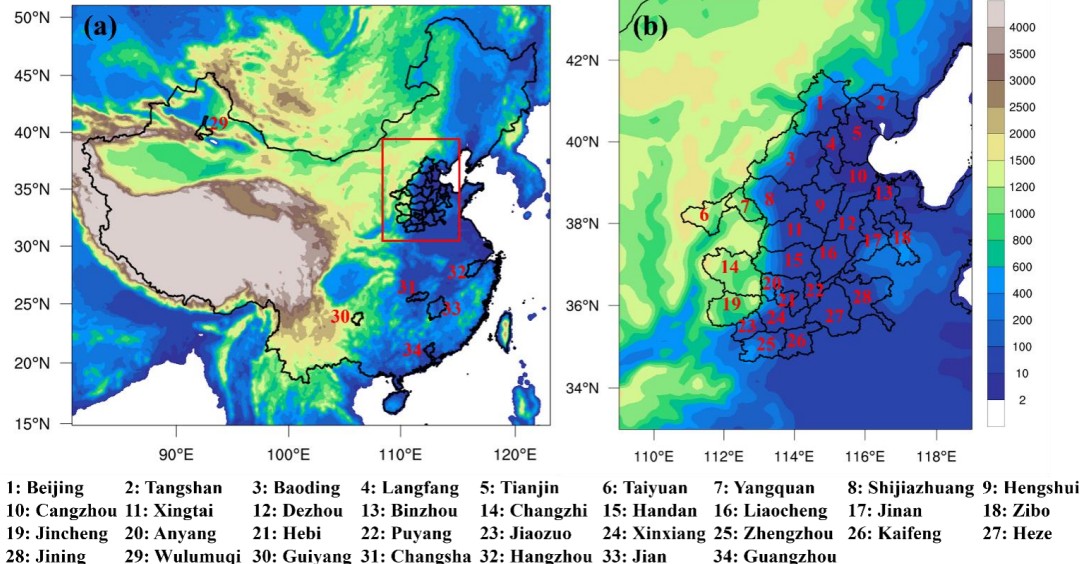

| 1: Beijing | 2: Tangshan | 3: Baoding | 4: Langfang | 5: Tianjin | 6: Taiyuan | 7: Yangquan | 8: Shijiazhuang | 9: Hengshui |
| 10: Cangzhou | 11: Xingtai | 12: Dezhou | 13: Binzhou | 14: Changzhi | 15: Handan | 16: Liaocheng | 17: Jinan | 18: Zibo |
| 19: Jincheng | 20: Anyang | 21: Hebi | 22: Puyang | 23: Jiaozuo | 24: Xinxiang | 25: Zhengzhou | 26: Kaifeng | 27: Heze |
| 28: Jining | 29: Wulumuqi | 30: Guiyang | 31: Changsha | 32: Hangzhou | 33: Jian | 34: Guangzhou | | |

Figure 1. (a) Map of the modeling domain and location of each target city in model evaluation. (b) The locations of BTH2+26 cities, denoted as the red frame in (a).



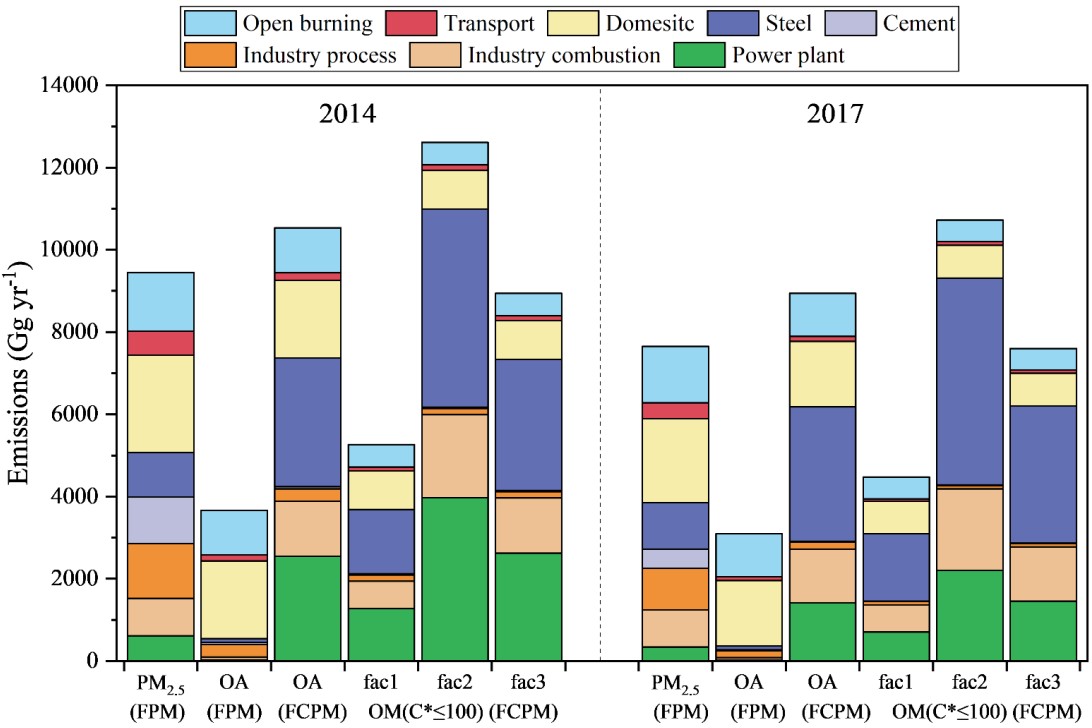

Figure 2. Annual emissions of PM$_{2.5}$ and OA in filterable particulate matter (FPM), OA in filterable plus condensable particulate matter (FCPM) before the volatility distributions, and OM (C*≤100) in FCPM after application of the volatility distributions for the fac1, fac2 and fac3 cases over China in 2014 and 2017.



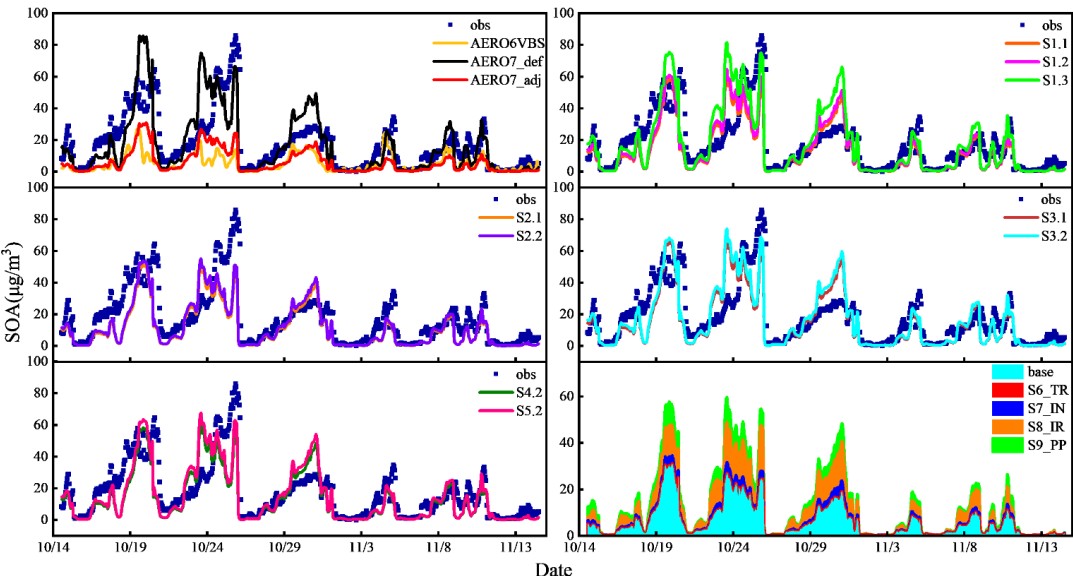

Figure 3. The observed and simulated hourly SOA concentrations during the episode from October 14 to November 14, 2014 at the Beijing site in the sensitivity cases as summarized in Table 3.



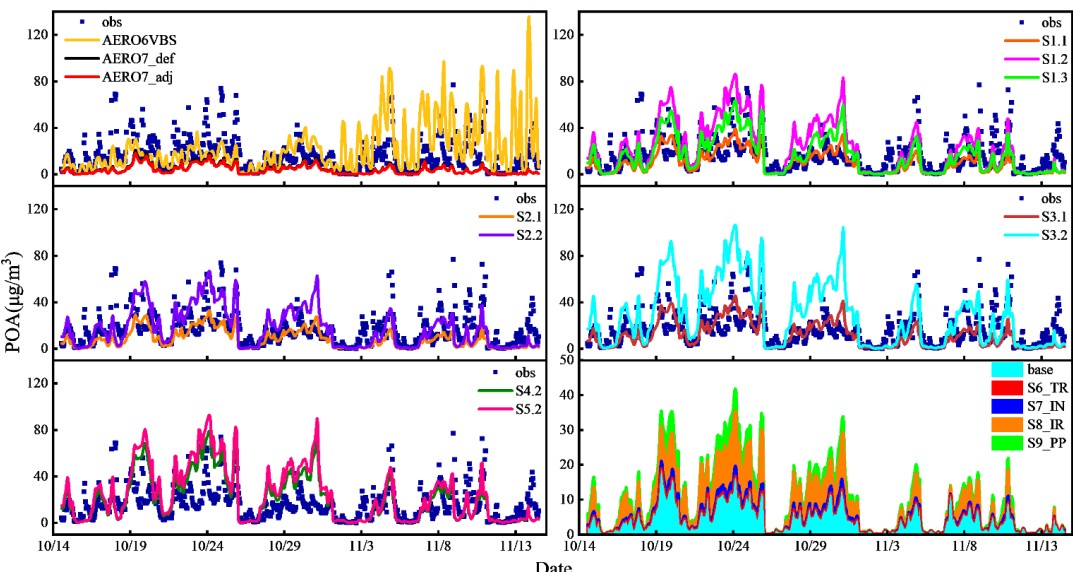

Figure 4. The observed and simulated hourly POA concentrations during the episode from October 14 to November 14, 2014 at the Beijing site in the sensitivity cases as summarized in Table 3.





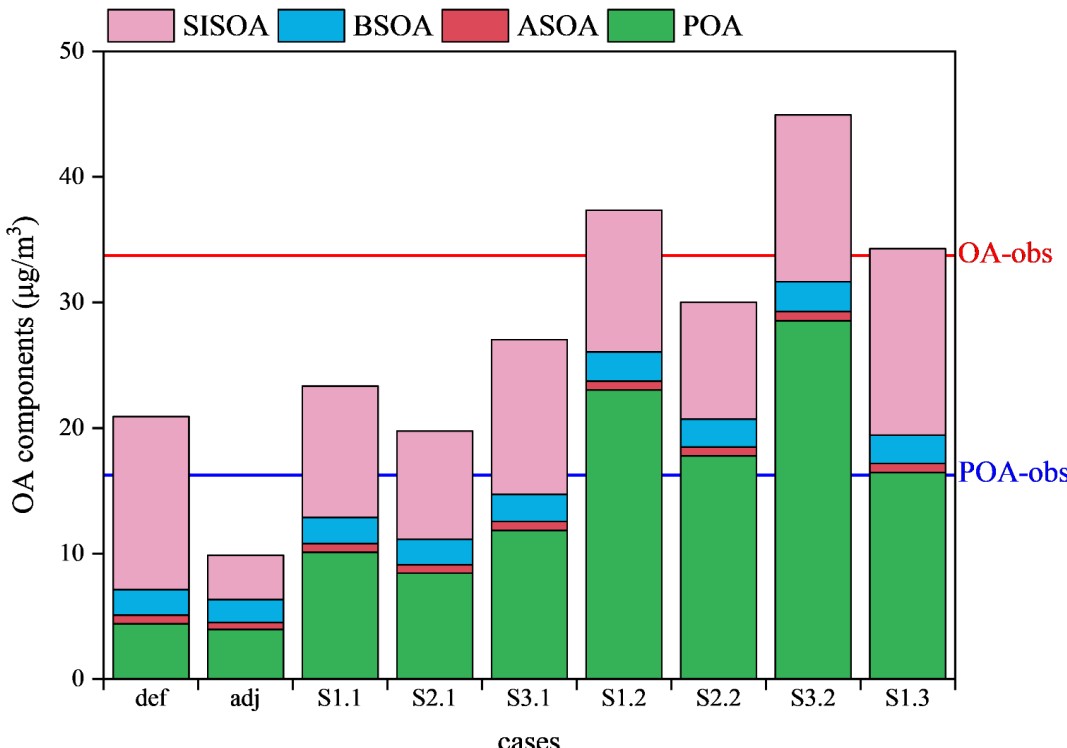


Figure 5. The simulation concentrations of different OA components averaged over the whole study period from October 14 to November 14, 2014 at the Beijing site in the sensitivity cases. AERO7_def is abbreviated as def and AERO7_adj as adj. ASOA, BSOA and SISOA denote SOA generated by anthropogenic VOCs, biogenic VOCs and low volatile S/IVOCs, respectively. The red and blue horizontal line denote the average observation concentrations of OA and POA, respectively.




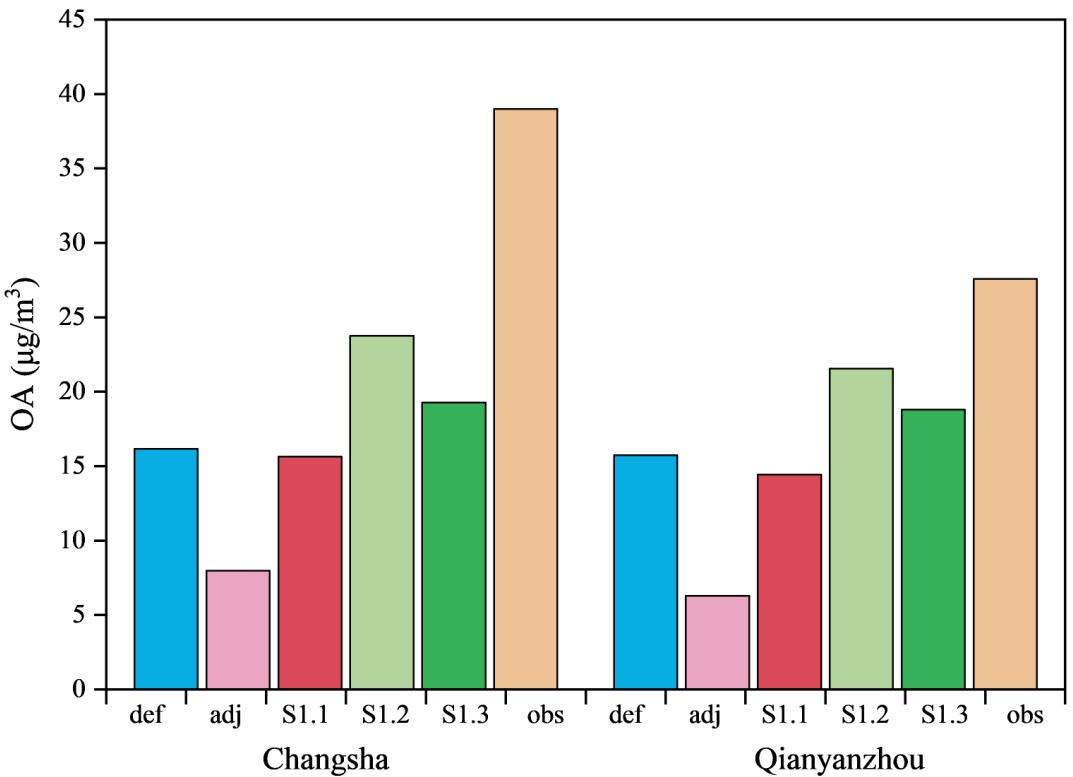

Figure 6. The observed and simulated OA concentrations in the sensitivity cases on November 3, 2014
at Changsha and Qianyanzhou.



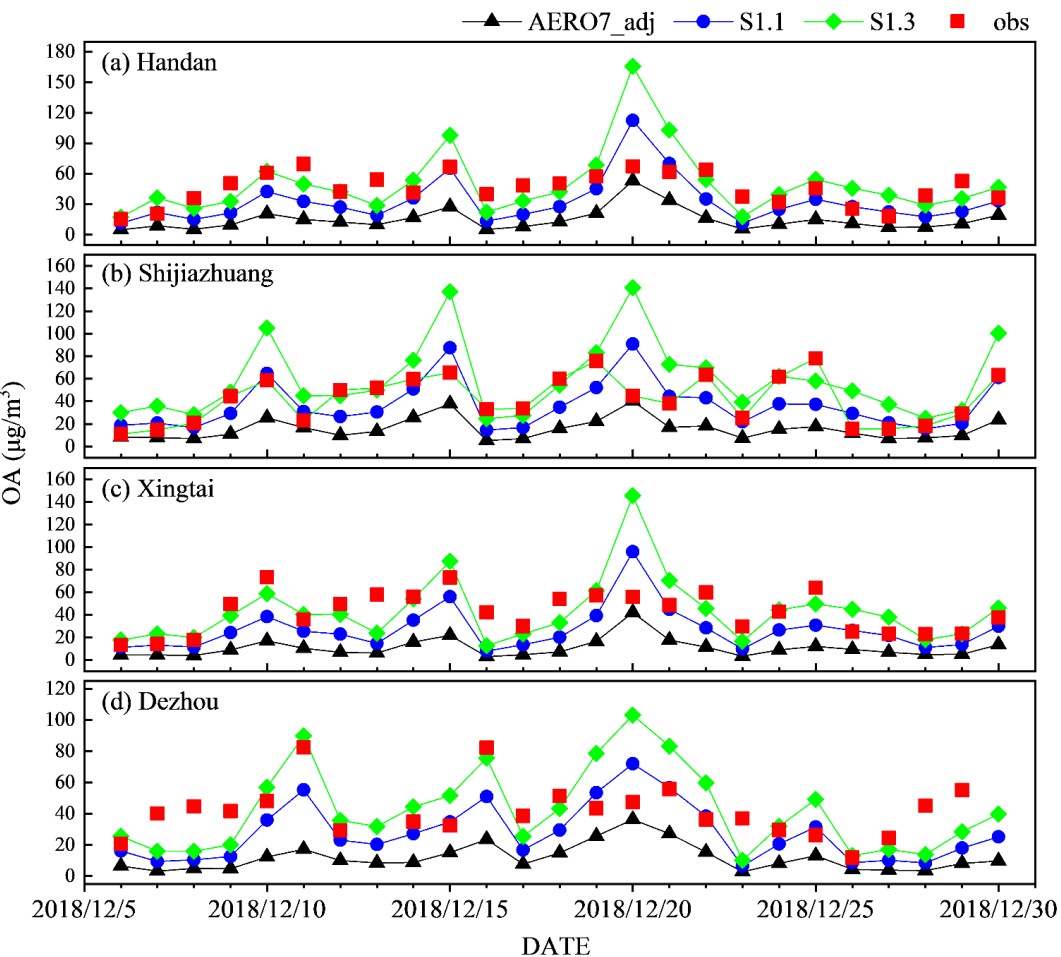

Figure 7. The observed and simulated daily OA concentrations during December 6-30 in 2018 at (a) Handan, (b) Shijiazhuang, (c) Xingtai and (d) Dezhou.

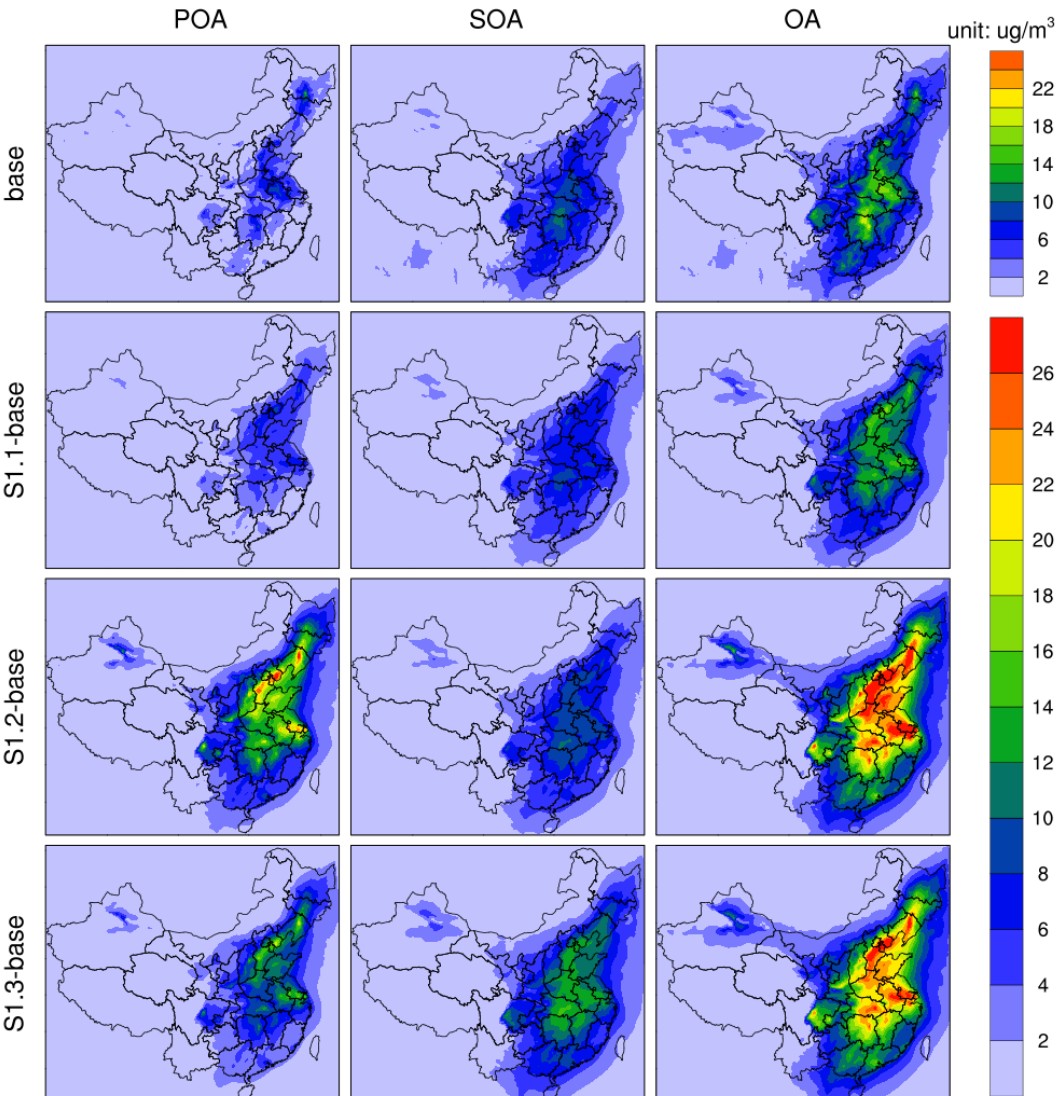

Figure 8. Spatial distributions of the concentrations of POA, SOA and OA averaged over the whole period of October 14-November 14 in 2014 generated by the simulations with FPM sources (base) and CPM sources (S1.1-base, S1.2-base, S1.3-base).





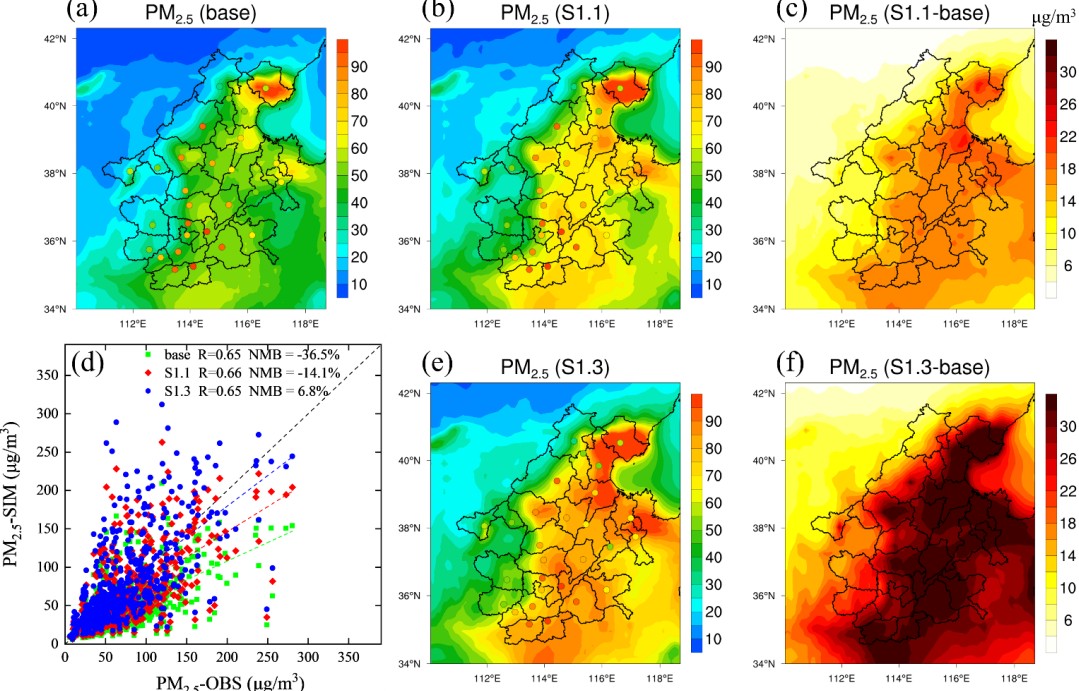

Figure 9. Spatial distributions of the average PM$_{2.5}$ concentrations during December 6-30, 2018, over the BTH2+26 cities in (a) base, (b) S1.1, (e) S1.3, (c) absolute difference between S1.1 and base, and (f) absolute difference between S1.3 and base. Among them, the PM$_{2.5}$ concentrations from December 22 to 26 are not included due to the missing observation data. (d) Scatter plots and linear regressions of observed (OBS) and simulated (SIM) daily PM$_{2.5}$ concentrations for all of the BTH2+26 cities during the above time period under the base, S1.1, and S1.3 scenarios.



Figure 10. (a) Spatial distributions of hourly $PM_{2.5}$ concentrations at some peak hours over the BTH2+26 cities under the base, S1.1, and S1.3 scenarios. The colored dots denote observation values for each city. (b) Scatter plots and linear regressions of observed (OBS) and simulated (SIM) hourly $PM_{2.5}$ concentrations for all cities under the base, S1.1, and S1.3 scenarios.