# Peer review of "Impacts of condensable particulate matter on atmospheric organic aerosols and fine"

_Atmospheric Chemistry and Physics, 2022_

## Author Comment (AC1)

**Reply to comments on "Impacts of condensable particulate matter on atmospheric organic aerosols and fine particulate matter (PM$_{2.5}$) in China" by Mengying Li et al.**

5 **We thank you for all the constructive comments and suggestions. We have adopted all of the suggestions in our revised manuscript. The followings are our point-to-point responses to the reviewer's comments. The responses are shown in brown and bold fonts, and the added/rewritten parts for the revision are presented in blue and bold fonts.**

**Reply to Reviewer #1**

10 Li et al. constructed an emission inventory for condensable particulate matter for China and evaluated its impact on the simulation of organic aerosol and PM$_{2.5}$. The study provides useful information on how CPM emissions (which are conventionally not measured in emission studies) affect ambient concentrations. However, I find the paper difficult to follow mainly because the method descriptions are not well organized. For example, the authors did not explicitly state their operational definitions for OA,
15 POA, SOA, FPM, CPM, OMlsi, OM(C*<100), SVOC, and their relationships, which cause confusion. For example, I am confused about whether CPM emissions are accounted for as only primary emissions or also as secondary emissions in the emission inventory? And whether this inventory assumes that all CPM are organic? Whether E_OA include E_OMlsi (Eq.1-3) or do they represent non-overlapping components? Clearly describing what the authors actually did and meant would definitely help assess the
20 scientific value of this study.

**Response: We truly appreciate all the constructive comments and suggestions of the reviewer. We have adopted all the suggestions in our revised manuscript. We clearly recognized that the methods descriptions were not well organized and might cause confusions. Therefore, we have revised the descriptions in the Materials and methods section and also supplemented a table to explicitly state**
25 **our operational definitions for OA, POA, SOA, FPM, CPM, OM$_{lsi}$, OM(C*<100), SVOC, and other used acronyms (see Table 1). The serial numbers of other tables have been adjusted. This will help readers better understand what we actually did and meant in this study.**
**For these points of confusions, our answers are as follows: CPM emissions are accounted for only primary emissions in the emission inventory; This study only concentrated on organic CPM**
30 **emissions, so the constructed inventory only included the emissions of organic CPM; In Eq.1-3, E_OA included E_OM$_{si}$, and E_OM$_{lsi}$ has been revised to E_OM$_{si}$.**

[revised manuscript text omitted]

**Minor comments:**

75   1. Line 76: ambiguous meaning of "negative impact". Change to "negative radiative forcing".
**Response: We thank the reviewer for the suggestion. "a negative impact on radiative climate forcing, air quality and human health" has been changed to "a negative radiative forcing and adverse impacts on air quality and human health" in the revision.**

2. Line 98: unclear what is "inapplicability of parameter localizations". Do you mean there is a lack of
80   local emission factors?
**Response: We thank the reviewer for the suggestion. "inapplicability of parameter localizations" means that the parameters which characterize SOA yields in models were not sufficiently localized, and not necessarily applied to all the regions. This has been changed to "lack of localized parameters" in the revised manuscript.**

85   3. Line 109: "totally" -> "completely"

**Response: We thank the reviewer for the suggestion. "totally" has been changed to "completely".**

4. Line 134: the ambiguous expression "more than 50% of organic composition were measured in CPM". Please rephrase.
**Response: We thank the reviewer for the suggestion. This has been rephrased to "CPM contained**
90   **more than 50% organic components".**

5. Line 148: "largely" -> "greatly" or "substantially"
**Response: We thank the reviewer for the suggestion. "largely" has been changed to "greatly".**

6. Table S3 should list the measurement methods that these studies used.
**Response: We thank the reviewer for the suggestion. The measurement methods have been listed**
95   **in the "test method" column of Table S3. The column name has been changed to "Measurement methods".**

7. Line 172 & Line 188-193: Please explain what is the difference between OA(CPM) and OM_lsi(CPM). Do you consider OA(CPM)-OM_lsi(CPM) as POA(CPM)?
**Response: We thank the reviewer for the suggestion. OA(CPM) denotes the primary organic matter**
100   **measured in CPM. To differentiate organic CPM emissions from simulated OA concentrations, OA(CPM) has been changed to OM(CPM) as shown in the supplemented Table 1. $OM_{lsi}$ (CPM) has been changed to $OM_{si}$ (CPM) in the revised manuscript. $OM_{si}$ (CPM) denotes a collective term for a range of organic matter with different volatility in CPM, containing organic matter which was semi-volatile (SVOC, $10^0 \leq C^* \leq 10^3$ µg m$^{-3}$), or had intermediate volatility (IVOC, $10^3 < C^* \leq 10^6$ µg**

m$^{-3}$). Thus OM (CPM) included OM$_{si}$ (CPM). Since the volatility characteristics of organic CPM have not been accurately determined in relevant measurement studies, the emissions of OM$_{si}$ (CPM) were scaled to OM (CPM) emissions in this estimate (see Eq.1-3), that is, the total emissions of OM (CPM) were distributed in different volatility bins.

8. Line 196: what is the relative importance of stationary combustion vs. vehicles?

Response: We thank the reviewer for the suggestion. As shown in Fig. 2-4, the emissions of organic CPM from stationary combustion sources and the contributions of CPM from these sources to atmospheric OA concentrations were much greater than that from vehicle sources.

9. Line 216-217 & 270-273: what are the bases of these scaling factors? I thought you already derived emissions based on Eq.1-3. Why do you need to scale with respect to POA emissions?

Response: We thank the reviewer for the suggestion. Although the POA emissions from CPM has been derived based on Eq.1-2, the emissions of organic matter with different volatility cannot be acquired on the basis of the current measurement results. Therefore, a list of scaling factors should be applied to POA emissions to calculate the emission rates of these organic matter, then to represent model species in CMAQv5.32. These mixed species underwent gas-particle partitioning and multi-generational gas-phase photochemical oxidations of organic vapors by OH radicals to generate successively lower volatility and more-oxygenated species, and then condensed to produce SOA.

10. Line 318-319: C*<=100 or C*<=10? Also state the unit of C*.

Response: We thank the reviewer for the suggestion. It should be C*<=100. The unit of C* is µg m$^{-3}$ and has been stated in the revised manuscript.

11. Line 322: The OA emissions reported here are for what geographical region?

Response: We thank the reviewer for the suggestion. The OA emissions reported here are for the whole China. It has been changed to "the emissions of OA over mainland China were 3664.6 Gg" in the revision.

12. Section 3.2. These evaluations are not relevant and are just distractive. I'd suggest removing the section or putting it in the supplementary material.

Response: We thank the reviewer for the suggestion. The Sect. 3.2 has been moved to the supplementary material.

13. Line 369-372. Use "episode" instead of "process".

Response: We thank the reviewer for the suggestion. "process" has been changed to "episode" in the revision.

14. Section 3.3. How do the observations distinguish POA and SOA? Their operational definition should be introduced, as sometimes it is not so straightforward to compare to simulations.

Response: We thank the reviewer for the suggestion. We have supplemented the operational definition to distinguish between POA and SOA in Sect. 2.4.

Added/rewritten part in Sect. 2.4 Observational data: To distinguish between SOA and POA, Aerosol Mass Spectrometer (AMS) measurements and the method of Positive Matrix Factorization (PMF) were used by Xu et al. (2015), identifying three POA factors from coal combustion, biomass burning and cooking, and two SOA factors of semi-volatile and low-volatility oxygenated OA.

145 15. Line 461-462: Why do you think it is more likely due to meteorological factors, rather than that your emissions are still underestimated and that there are still missing SOA pathways?

**Response: We thank the reviewer for the good suggestion. Indeed, we ignored the influence of other factors. This has been changed to "the underestimation of our estimated CPM emissions, effects of meteorological factors and other missing SOA formation pathways" in the revision.**

150 16. Figure 1. Explain in the caption what the color shading represents

**Response: We thank the reviewer for the suggestion. The color shading represents the regional altitude. We have supplemented the explanation in Fig.1 and caption in the revision.**

**Added/rewritten part in Figure 1:** Figure 1. (a) Map of the modeling domain and location of each target city
155 in model evaluation. (b) The locations of BTH2+26 cities, denoted as the red frame in (a). The color shading represents the regional altitude.

---

## Author Comment (AC2)

**Reply to comments on "Impacts of condensable particulate matter on atmospheric organic aerosols and fine particulate matter (PM$_{2.5}$) in China" by Mengying Li et al.**

We thank you for all the constructive comments and suggestions. We have adopted all of the suggestions in our revised manuscript. The followings are our point-to-point responses to the reviewer's comments. The responses are shown in brown and bold fonts, and the added/rewritten parts for the revision are presented in blue and bold fonts.

**Reply to Reviewer #2**

Organic aerosol, especially secondary organic aerosol (SOA), is a major component of the overall aerosol loading in various environments around the globe and significantly influences the air quality and climate. However, there is a significant gap between observed and modeled SOA. One possible reason is the incomplete information on emissions and properties of SOA precursors. This work modified the aerosol emission inventory by including condensable particulate matter (CPM) and tested this inventory by simulating some observations in Beijing and Beijing-Tianjin-Hebei region. The manuscript is overall well-written and fits the scope well of ACP. However, I have a few comments to be addressed before this manuscript can be published.

**Response: We truly appreciate all the constructive comments and suggestions of the reviewer. We have adopted all the suggestions in our revised manuscript.**

**1. Estimations of CPM emissions (Line 160 – 198):** It is reasonable to assume that CPM contains many condensable substances, such as nitric acid, ammonia, S/IVOCs, and the gas-particle partitioning of which is regulated by temperature. Therefore, have many of the CPM components already been considered in the gas emissions inventory? This may lead to repeated consideration of many compounds and further overvaluing of emission intensity; please clarify how organic (gas + particle) emissions were considered in this study?

**Response: We thank the reviewer for the good suggestion. Since the current measurement methods for PM in stationary exhaust sources in China (GB/T 16157-1996) have only involved the collections of FPM, the CPM components have not been considered in the emission inventory. First, for inorganic gases, the gas emission inventory contained SO$_2$, NOx, CO and NH$_3$, thus the condensable substances in CPM, such as nitric acid and ammonia have not been included in the current emission inventory. Then for organic gases, the current gas emission inventory only contained gaseous NMVOCs directly emitted by the exhaust sources, which was different from the CPM components. For organic (gas + particle) emissions in this study, the organic CPM is a mixture of components which is semi-volatile (SVOCs, $10^0 \leqslant C^* \leqslant 10^3$ μg m$^{-3}$), or has intermediate volatility (IVOCs, $10^3 < C^* \leqslant 10^6$ μg m$^{-3}$), thus it is definitely different from the organic gases with higher volatility ($C^* \geqslant 10^7$ μg m$^{-3}$) in the current emission inventory. Considering that organic FPM from the stationary combustion and mobile sources mainly contained low volatile matter, so all of these emissions were**

assigned to the LVPO1 species in the CMAQ. Then the organic CPM components were assigned to the CMAQ species of different volatility bins (LVPO1, SVPO1, SVPO2, SVPO3 and IVPO1). As mentioned in Sect. 2.1, organic CPM is composed of organic matter which is semi-volatile or has intermediate volatility, thus the first bin which represents nonvolatile organic matter should be set to zero. These mixed species underwent gas-particle partitioning and multi-generational gas-phase photochemical oxidations to generate SOA. Therefore, the inclusion of organic CPM did not lead to repeated consideration of many compounds and overvaluing of emission intensity.

**2. The model configuration (Line 200 – 246):** The emitted CPM can be further oxidized in the atmosphere to produce SOA. How does the model consider this process? It seems there is no component information of the CPM. Therefore, it is not clear to me how CPM performs atmospheric chemical reactions.

**Response: We thank the reviewer for the suggestion. As stated in the Sect. 4 Conclusions, one of the limitations in this study was that there was no explicit component information and volatility characterization of primary organic CPM species available. Thus the organic CPM emissions were lumped into the original surrogate volatility species (LVPO1, SVPO1, SVPO2, SVPO3, and IVPO1) in the CMAQ model for representing the SOA formation from CPM. These mixed species underwent gas-particle partitioning and multi-generational gas-phase photochemical oxidations of organic vapors by OH radicals to generate successively lower volatility and more-oxygenated species, and then produce SOA.**

**Added/rewritten part in Sect. 2.3  Design of sensitivity simulation cases:** the emissions of organic CPM were mapped to surrogate species for different volatility bins (LVPO1, SVPO1, SVPO2, SVPO3, and IVPO1) in the CMAQ model for representing the SOA formation from CPM. These mixed species underwent gas-particle partitioning and multi-generational gas-phase photochemical oxidation of organic vapors by OH radicals to generate successively lower volatility and more-oxygenated species, and produce SOA.

**3. Line 241:** Please provide the reasons for setting the emissions to be reduced by 30%.
**Response: We thank the reviewer for the suggestion. We have provided the reasons for setting the emissions to be reduced by 30% in the revision.**

**Added/rewritten part in Sect. 2.2 The model configuration:** Based on the observed reductions in the concentrations of PM$_{2.5}$, SO$_2$, NO$_2$, and CO during APEC in Beijing and its surrounding cities (Li et al., 2017e, 2019; Wen et al., 2016), and 28% contribution of the emission control measures to the reduction of PM$_{2.5}$ concentrations (Liang et al., 2017), thus the approximate emission reduction of 30% was conducted during the above time period for the region with two municipalities (Beijing and Tianjin) and five provinces (Hebei, Shanxi, Henan, Shandong, and Inner Mongolia).

**4. Table 4:** This study can be seen as an improvement to the emission inventory, but it seems that the model simulates SOA better than POA. Please clarify this.
**Response: We thank the reviewer for the good question. Indeed, organic CPM contributes to the primary emissions of OA in the emission inventory, but it is composed of organic matter which is semi-volatile (SVOCs, $10^0 \leqslant C^* \leqslant 10^3$ μg m$^{-3}$), or has intermediate volatility (IVOCs, $10^3 < C^* \leqslant 10^6$ μg m$^{-3}$), and it can go through the gas-particle partitioning and aging reactions to generate SOA. Therefore, the organic CPM can contribute to both POA and SOA. Based on better simulations of SOA than POA, it further suggests that CPM has a greater impact on SOA than POA.**

**5. Line 338-339:** how to make the estimation of uncertainties?

**Response: We thank the reviewer for the suggestion. The estimation of uncertainties related to variabilities in the ratio of $E_{OM}$(CPM) to $E_{PM2.5}$(FPM) was described in Sect. 2.3 as follows "We carried out bootstrapping and Monte Carlo simulations to obtain the mean and uncertainty ranges of $E_{OM}$(CPM)/$E_{PM2.5}$(FPM) for stationary combustion sources including power plant (PP), industry combustion (IN), steel (IR) (see Table 3). First, the optimal probabilistic distributions and uncertainty ranges were determined for each source category. Then the statistical bootstrap simulation was applied to calculate the mean and 95% confidence interval of emission ratios for each source category. Finally, the uncertainties of these parameters were propagated to calculate the total uncertainty of emission by running Monte Carlo simulations for 10,000 times. Notably, the estimated uncertainties were only related to variabilities in the ratio of $E_{OM(CPM)}$ to $E_{PM2.5}$(FPM), but did not necessarily represent the overall uncertainties of organic CPM emissions."**

**6. Line 82:** the reference here should be Huang et al., 2014.

**Response: We thank the reviewer for the suggestion. We have revised Huang et al. (2015) to Huang et al. (2014).**

7. The distinction between SOA and POA based on OC/EC measurements is debatable. I recommended using some AMS data, published a lot during the last decades in the Beijing-Tianjin-Hebei region, to validate.

**Response: We thank the reviewer for the suggestion. Actually, the observation data of OA in 2014 was measured by an Aerodyne high-resolution time-of-flight aerosol mass spectrometer (HR-AMS). Then three POA factors and two SOA factors were identified using positive matrix factorization (PMF) with the AMS measurement data. Therefore, the SOA and POA data were not based on OC/EC measurements, and can be used to validate the simulation results. This information has been added to the Sect. 2.4 Observational data.**

**Added/rewritten part in Sect. 2.4 Observational data:** To distinguish between SOA and POA, Aerosol Mass Spectrometer (AMS) measurements and the method of Positive Matrix Factorization (PMF) used by Xu et al. (2015) identified three POA factors from coal combustion, biomass burning and cooking, and two SOA factors of semi-volatile and low-volatility oxygenated OA used in this study.